# Pathway centric analysis for single-cell RNA-seq and spatial transcriptomics data with GSDensity

Qingnan Liang[1], Yuefan Huang[1], Shan He[1] & Ken Chen [1]✉

Advances in single-cell technology have enabled molecular dissection of heterogeneous biospecimens at unprecedented scales and resolutions. Cluster-centric approaches are widely applied in analyzing single-cell data, however they have limited power in dissecting and interpreting highly heterogenous, dynamically evolving data. Here, we present GSDensity, a graph-modeling approach that allows users to obtain pathway-centric interpretation and dissection of single-cell and spatial transcriptomics (ST) data without performing clustering. Using pathway gene sets, we show that GSDensity can accurately detect biologically distinct cells and reveal novel cell-pathway associations ignored by existing methods. Moreover, GSDensity, combined with trajectory analysis can identify curated pathways that are active at various stages of mouse brain development. Finally, GSDensity can identify spatially relevant pathways in mouse brains and human tumors including those following high-order organizational patterns in the ST data. Particularly, we create a pan-cancer ST map revealing spatially relevant and recurrently active pathways across six different tumor types.

scRNA-seq methods have been widely applied to delineate cellular-molecular heterogeneity of tissues, with novel cell types and cell states uncovered in many different contexts[1-5]. The current practice of scRNA-seq analysis usually split cells into discrete clusters[6] and then focus on one or a few clusters of interest to study their biological meanings, which is herein referred to as 'cell-centric' data analysis (Fig. 1a, top). However, clustering process itself becomes less reliable when samples contain cells under active state transition, which is a common situation in tumor or developmental datasets[3,7-9]. In such cases, the performance of cluster-level pathway analysis methods, such as gene ontology (GO) enrichment analysis, are hampered by unreliable clustering results. Moreover, the list of pathways obtained from these analyses is often fairly long, limited by gene sets and annotations available in the databases, and the statistical P-values are often difficult to interpret. In contrast, for a given single-cell dataset, nominating pathways of interest is often more straightforward than nominating cell subpopulations of interest, because the former could be more easily obtained from biological domain knowledge, or functional screen experiments such as

those in DepMap (https://depmap.org/portal/). Thus, given a single-cell dataset, focusing directly on pathways of interest in a cluster-independent manner, or 'pathway-centric' data analysis (Fig. 1b, bottom), could be a powerful, alternative approach in single-cell analysis to generate highly interpretable results.

Successful pathway-centric analysis of single-cell data should contain at least two functions: first, it should distinguish whether a pathway, in the format of a gene set, is truly heterogenous among the cells without clustering them; second, it should be able to accurately evaluate pathway activity at single-cell level and fetch the most relevant cells for downstream analysis. Some computational tools are available for evaluating single-cell level activity or enrichment of a given gene set (the second function), such as single-sample gene set enrichment analysis (ssGSEA[10]), AUCell[11], and CelliD[12], etc., but they all lack an overall evaluation of pathway heterogeneity among the cells (the first function). Also, some of these methods are susceptible to sparsity and technical noise in single-cell data. Matrix factorization-based tools, on the other hand, can partition genes to a

[1]Department of Bioinformatics and Computational Biology, UT MD Anderson Cancer Center, Houston, TX, USA. ✉e-mail: kchen3@mdanderson.org

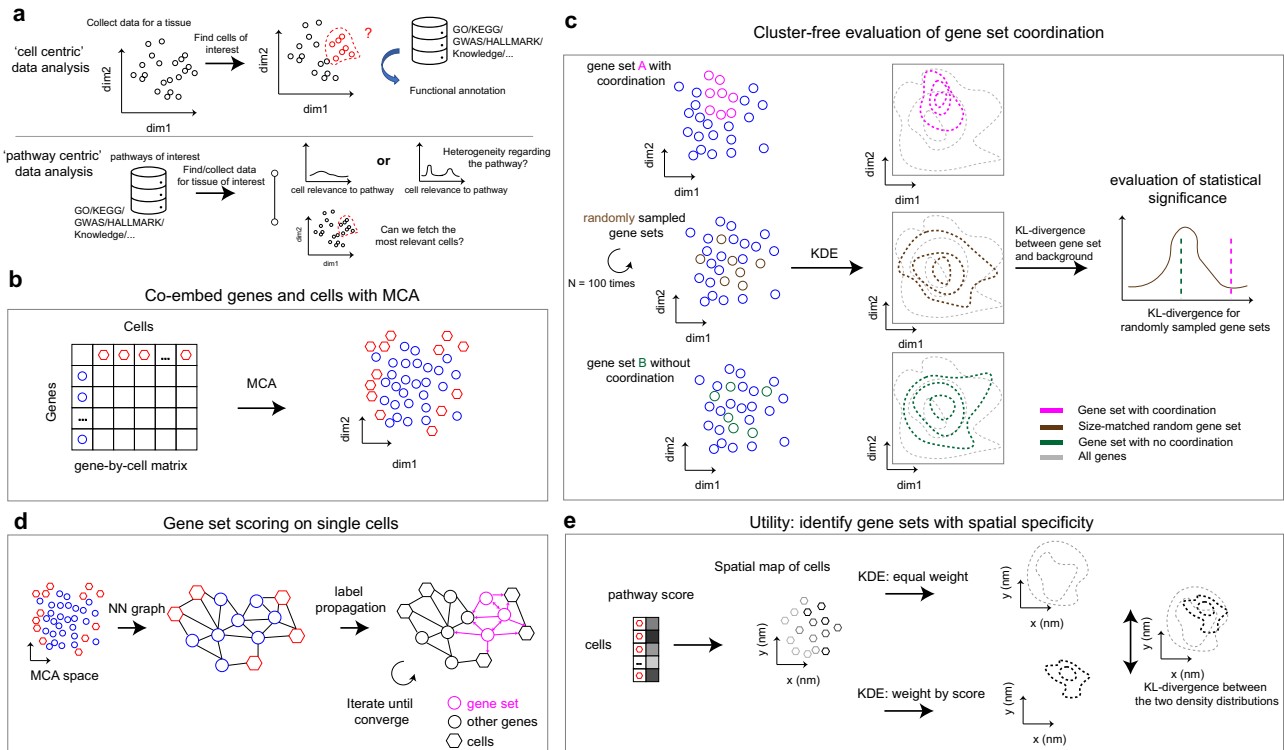

**Fig. 1 | Overview of the GSDensity method. a** GSDensity allows for 'pathway centric' analysis of scRNA-seq data. Current analysis approaches often focus on a cell population of interest and apply pathway analysis to functionally annotate the population (top). GSDensity allows for cluster-free testing of whether genes from the same pathway display coordination towards a cell population, and then fetch the cell population. This allows for direct investigation of whether a pathway of interest is heterogenous in a scRNA-seq data (bottom). **b** GSDensity first projects genes and cells to the same low-dimensional embeddings using MCA. **c** For each gene set (gene set A and B in the top and bottom panel), the density in the MCA space is estimated and compared with the density of all genes (gray contour) using

KL-divergence as the metric. Size-matched random gene sets are used to evaluate the significance of the KL-divergence. Gene sets with coordination have higher KL-divergence. **d** A nearest-neighbor graph containing both cells and genes is constructed utilizing the MCA embedding. The genes in the gene set of interest are treated as 'seeds' for label propagation. This approach is to evaluate the relevance of each cell towards the gene set. **e** To evaluate the spatial relevance of a gene set, we perform two kernel density estimations on the two-dimensional spatial map with each cell having equal weights or using the pathway scores as the cell weights, respectively. The KL-divergence between these two density distributions is used to evaluate the spatial relevance of the gene set.

set of 'factors' (or 'patterns'), and these factors usually demonstrate certain levels of heterogeneity among the cells[13]. However, construction of these factors are mostly data driven, lack of input from knowledgebases, and thus the functional interpretation of these factors still relies on manual annotation using tools such as GO enrichment analysis.

Here, we present a computational framework, GSDensity, for pathway-centric analysis of single-cell and ST data. GSDensity uses multiple correspondence analysis[14] (MCA) to co-embed cells and genes into a latent space and quantifies the overall variation of pathway activity levels across cells by estimating the density of the pathway genes in the latent space. Pathway activity for each cell can be calculated using network propagation in a nearest-neighbor cell-gene graph, with pathway genes used as seeds for random walks. When spatial information of cells is available (e.g., in ST datasets), the spatial relevance of a pathway is reflected by the density of cells in the two-dimensional image, weighted by their pathway activity scores. Through validation and benchmarking on multiple real and simulated datasets, we prove that GSDensity is capable of distinguishing truly heterogenous gene sets and inferring pathway activities in each cell with superior accuracy, compared to six widely applied frameworks. We demonstrate the usage of GSDensity in cluster-free, pathway-based classification of tumor cells and found an association between the GAS6-TYRO3 signaling and tumor cell division in multiple independently collected triple negative breast cancer samples. We also show how GSDensity can be used in conjunction with trajectory analysis

tools to group signaling pathways by their activity patterns over various developmental stages. Finally, we use three examples to show how GSDensity can identify spatially related pathways in the context of brain functions and of immune infiltration of different tumor types.

## Results

### An overview of the GSDensity method

In the context of scRNA-seq data, considering a curated functional gene set, when the genes from this gene set are highly and specifically expressed in a subpopulation of the cells, we call such a gene set 'coordinated'. This subpopulation of cells is thus defined as highly relevant cells to this gene set. A randomly selected list of genes is anticipated to be of low coordination, with no relevant cells. GSDensity is designed to detect this coordination of a gene set without having to partition cells beforehand. GSDensity first projects cells and genes in scRNA-seq data into the same low dimensional space using MCA (Fig. 1b), inspired by a previous study, CelliD[12]. In the MCA space, distances among cells, genes, or between cells and genes reflect their association. Thus, we hypothesize that when a gene set has true coordination, the genes would appear clustered in the MCA space, occupying a relatively small subspace (Fig. 1c, top). In contrast, they would appear randomly distributed when they are not coordinated (Fig. 1c, bottom). The level of gene set coordination can be measured by contrasting the density of the pathway genes and that of all the genes using KL-divergence. The density level of a gene set is calculated using kernel density estimation (Gaussian kernel) in the MCA space.

Randomization is used to estimate how likely a differential density level is reached by chance. We randomly sample multiple size-matched gene sets and compute differential density in the same way (Fig. 1c, middle), and the resulting differential density levels are used to generate a null distribution for estimating statistical significance (Methods). This whole process does not require any information of cell clustering or annotations.

Calculating pathway activity levels (PALs) in each cell is a critical step for fetching the cells most relevant for a pathway. GSDensity first constructs a nearest-neighbor graph based on the projected distances between the cells and the genes in the co-embedding MCA space. Using a low-dimensional embedding generated by MCA is beneficial for alleviating the effect of sparsity and technical noise in single-cell data. For a gene set, GSDensity calculates the PAL of each cell, by performing random walk with restart on the graph using pathway genes as seeds (Fig. 1d). After convergence, each node (both cell and gene) will have a score, reflecting its relevance to the pathway. For single-cell pathway activity analysis, we normalize the PAL scores across cells and further split the cells into two groups with high and low relevance to the pathway by binarizing the PAL scores with the antimode.

We then extend GSDensity to evaluate whether a pathway is spatially related in a ST data. The single-cell PALs are calculated the same way as described above, and the activity scores are then used as weights for cells to calculate a weighted kernel density in the two-dimensional spatial map. We also calculate a reference weighted kernel density with each cell having an equal weight, reciprocal to the number of cells. The difference of these two density measurements could then reflect the level of spatial relevance of a pathway, again quantified by the KL-divergence between the density distributions (Fig. 1e). For each pathway, we use label shuffling to generate a set of density distributions as controls for random chance.

## Validation and benchmarking for GSDensity

We first evaluate whether GSDensity can distinguish gene sets with true coordination from those with no coordination. We would like to use the peripheral blood mononuclear cells (PBMC) dataset for illustration. This dataset was curated and annotated by the SeuratData program ('pbmc3k' dataset; Supplementary Fig. 1a). For illustration, let us focus on the B cell marker genes. After co-embedding genes and cells (Supplementary Fig. 1b), we visualized the density of all genes, B cell markers, size-reduced B cell markers with three-folds random genes, and size-matched random genes (Fig. 2a). B cell markers displayed a compact distribution in the space which appeared very different from that of all genes. On the other hand, the random gene set showed a relatively similar density with that of all genes. The size-reduced markers with three-folds random genes showed some differences to all genes, but not as compact as the gene set with all markers. This result intuitively demonstrated the feasibility of using density to measure whether a gene set has true coordination, without focusing on any cell clusters at the first place. For a more systematic validation, we collected 8 real-world datasets (Table 1) and curated cell type marker genes from public databases (Methods): XCell[15], PanglaoDB[16], and CellMarker2[17]. Known marker gene sets were treated as ground-truth coordinated gene sets to validate GSDensity. We also created additional gene sets by mixing subsets of marker genes with random genes at various set sizes and proportions (Fig. 2b). We found that for all the 8 real-world datasets, the marker sets received the highest significance values at an alpha level of 0.05 (Fig. 2c–e, Supplementary Fig. 2), while the mixed sets resulted in lower significance and those with random genes showed almost no significance. For better demonstrating the performance of GSDensity, we showed the original p-value without adjustment. These results suggested that GSDensity can distinguish gene sets with true coordination from those with weak or no coordination. It is worth noting that the sizes of the marker gene sets vary from dozens to hundreds, which are in the same range of

most curated pathway gene sets, and thus indicating that GSDensity is reliable to be applied in practical pathway analysis settings.

We also wanted to test whether GSDensity performs well in scoring PALs on single cells and how it compares with other gene set scoring tools. We applied two metrics for assessing the performance of a method. First, given a cell type specific gene set, an optimal tool would assign high scores to cells from the corresponding cell type and low scores to other cells (Supplementary Fig. 1c). Such a specificity could thus be quantified using area under curve (AUC) in recovering the correct cell type at various score thresholds. Second, if a marker gene list were available for every cell type in a dataset, the PAL scores can be used to predict cell type identities: a cell is predicted to the type whose marker set has the highest score. Thus, an optimal tool would have the highest prediction accuracy (Supplementary Fig. 1d). We used both simulated scRNA-seq data and real-world data to perform the benchmarking experiments.

For simulation, we generated datasets from 3 modes, including the scenarios of stable, discrete cell types and dynamic, continuous cell states using SERGIO[18], which models transcriptional regulation of single-cell gene expression. We simulated dropout rate at three levels: 65%, 78%, and 90% zeros in the final expression matrices, in a range similar to what is often observed in real data. We were also able to define ground-truth gene sets at various levels of specificity, strong, medium, or weak, to a given cell type or state (Methods). For each mode, we included three repeats out of random initiation, and thus resulted in 27 simulated datasets in total (Table 2). SD (simulated data) 1-9 were from Mode-1 with three stable cell states. SD10-18 from Mode-2 with four continuous cell states under a bifurcation trajectory, and SD19-27 from Mode-3 with six continuous cell states with a trifurcation trajectory. We found that GSDensity can well distinguish ground-truth marker gene sets from random gene sets (Supplementary Fig. 3). We compared GSDensity with another six popular gene set scoring methods, including ssGSEA[10], GSVA[19], AUCell[11], VAM[20], scGSEA[21], and CelliD[12], using the two metrics introduced above. We found that GSDensity consistently outperform other methods in all the cases (Fig. 2f, Supplementary Fig. 4a–c). As expected, the performance of all methods decreased as the drop-out rate increased, with GSDensity being the least affected. For the simulated datasets, CelliD often achieved the second best performance, which implied the benefit of utilizing the MCA method in the PAL scoring task.

Similarly, we used the eight real-world datasets mentioned above to benchmark the PAL scoring of GSDensity and the other six methods. Since we did not have prior knowledges to construct marker gene sets with different specificity levels, we reduced the specificity of the marker sets by reducing the size and mixing them with randomly selected genes (Fig. 2g, Supplementary Fig. 4d, Supplementary Fig. 5a–e). Cell typing accuracy varied widely across datasets and gene sets. For example, the performance of all the methods were better in the dataset 'heart' (Supplementary Fig. 5c) than in the dataset 'lung' (Fig. 2g). Across the conditions, GSDensity outperformed most of the methods or at least achieved the top 3 position. We summarize the performance of PAL scoring methods in simulated and real-world datasets in Fig. 2h (simulated) and Fig. 2i (real-world), and we show that GSDensity has the most stable performance, as it shows more resistance to the increasing dataset sparsity or decreasing gene set specificity. In general, besides GSDensity, CelliD, scGSEA, and VAM showed reasonable performance.

## Classification of TNBC tumor cells using GSDensity

GSDensity allows examining any gene set, originated from biological insights or external studies, on a single-cell dataset, and thus enabling the utilization of domain knowledge to generate novel testable hypotheses and overcome limited data collection. Here we present an example. Cell proliferation or cycling has been known as a hallmark of cancer cells[22,23] and associated specifically to TNBC subtypes with poor

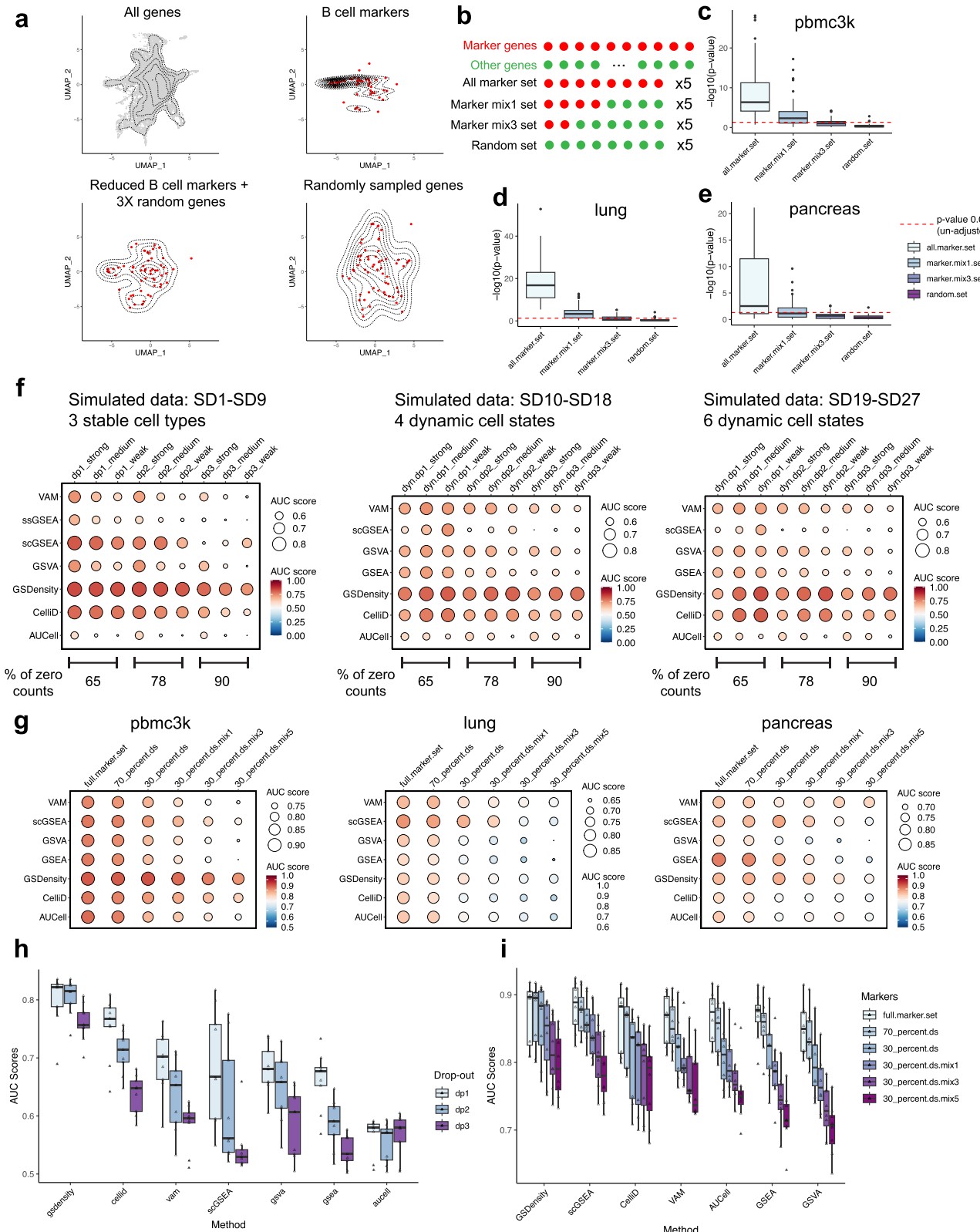

prognosis[24]. However, which subsets of cells are involved and how are proliferation sustained in the complex tumor immune microenvironment (TIME) are often unknown and difficult to hypothesize. We examined a scRNA-seq dataset[25] (TNBC-1) obtained from a triple-negative breast cancer (TNBC) patient. Since we are particularly interested in understanding why certain tumor cells are more proliferative and what are the associated TIME features, we used the

'hallmark G2M checkpoint' gene set[26] to classify tumor cells and fetch the actively dividing cells (Fig. 3a). The subset of cells also appeared associated with other hallmark features such as mitotic spindle, mTORC1 signaling and glycolysis (Fig. 3b–d), which are features indicative of actively dividing cells. This set of cells also showed higher expression of *MKI67* (encoding Ki-67) than other tumor cells (Fig. 3e, *p*-value < 2.2e−16, Wilcoxon test). The transcriptomes of tumor cells

**Fig. 2 | Benchmarking the GSDensity method. a** An illustration of gene set density in pbmc3k data using all genes (top left), B cell markers (top right), reduced B cell markers with random genes (bottom left) and randomly sampled genes (bottom right). It is worth notice that in the real data analysis, GSDensity directly estimate the density of gene sets in the MCA space, not the UMAP space. **b** Schematic of the gene sets used for panel **c–e. c–e** Validation of the sensitivity of GSDensity to identify gene sets with coordination. Marker sets generated following the strategy in **b** were used as input in the pbmc3k (**c**), lung (**d**), and pancreas (**e**) data, respectively. The red dashed line showed the unadjusted *p*-value equal to 0.05. One-sided t-test was used (Method). The center line of the box plot showed the median of data; the box limits showed the upper and lower quartiles; the whiskers showed 1.5 times interquartile range and points showed outliers. *n* = 40 for each box. Source data are provided as a Source Data file for panels **c–i. f** Benchmarking the reliability of gene set scoring aspect of GSDensity and six popular tools on simulated datasets. Each row represents a method, and each column represents the gene set and dataset condition. The colors and the sizes of the dots both demonstrate the AUC score. **g** Benchmarking the reliability of gene set scoring aspect of GSDensity and six popular tools on real datasets. Cell type markers were first used with their original sizes, and then got their specificity decreased by reducing the size to 70% and 30%, and further by mixing with random genes. **h** Summary of benchmarking experiments with simulated data using the AUC metric. For each simulated dataset, the median of the AUC scores were used as a data point in this boxplot. *n* = 9 for each box. **i** Summary of benchmarking experiments with real data using the AUC metric. For each simulated dataset, the median of the AUC scores were used as a data point in this boxplot. *n* = 8 for each box.

**Table 1 | Basic information for public, real-world datasets collected for benchmarking experiments**

| Data | # Genes | # Cells | # Cell types | Sparsity | Technology | Reference |
|---|---|---|---|---|---|---|
| pbmc3k | 11139 | 2638 | 8 | 92.42% | 10x Genomics | SeuratData |
| bmcite.small | 17009 | 9521 | 10 | 94.9% | RNA-seq data from CITE seq | SeuratData |
| pancreas | 15117 | 3390 | 9 | 88.41% | inDrop | Baron et al.[84] |
| liver.immune | 14550 | 5729 | 9 | 91.40% | 10x Genomics | Zhao et al.[85] |
| spleen.immune | 14452 | 4888 | 11 | 87.97% | 10x Genomics | Zhao et al.[85] |
| hcortex | 30046 | 2920 | 8 | 90.50% | Fluidigm C1 | Nowakowski et al.[86] |
| lung | 14483 | 7193 | 7 | 88.19% | 10x Genomics | Muus et al.[87] |
| heart | 15971 | 3474 | 7 | 87.94% | 10x Genomics | Koenig et al.[88] |

were highly affected by their CNV profiles, while interestingly, neither CNV-based nor transcriptome-based clustering would detect these cells as a group, using cluster-centric approaches. With CNV-based clustering (Supplementary Fig. 6a), the actively dividing cells appeared as disjoint subpopulations in all four tumor clones (Fig. 3f). With transcriptome-based clustering, the number of clusters were difficult to decide in this dataset (Fig. 3g). The degree of overlap between transcriptome-based clusters and the actively dividing cells was low (Supplementary Notes).

The proliferative ability of tumor cells could be regulated through their interactions with the TIME[27,28]. We then compared these actively dividing cells with other tumor cells from the angle of cell-cell interaction between tumor and TIME. We annotated the normal cells in the TNBC-1 dataset into two groups, fibroblasts, and immune cells (Supplementary Fig. 6b, c) and inferred the ligand-receptor interactions[29] between immune and tumor cells and between tumor and tumor cells. We displayed all the ligand receptor-pairs in Supplementary Fig. 7a and found that the dividing cells showed a distinct profile of the GAS6-TYRO3 axis. All the other ligand-receptor pairs showed differential enrichments in either immune-tumor interaction or tumor-tumor interaction, while GAS6-TYRO3 was the only pair that showed differential enrichments in both groups. We then found that both the tumor cells and the immune cells could express the ligand, GAS6, while only the dividing tumor cells displayed high expression of the receptor, TYRO3 (Fig. 3h), which indicated the specific activation of the TYRO3 downstream signaling in those cells. For confirmation, we also fetched actively dividing tumor cells from another two TNBC datasets[25], TNBC-2 and TNBC-5, and these cells also consistently showed high relevance to glycolysis, mTORC1 signaling, and mitotic spindle (Supplementary Fig. 6d–k, *p*-value < 2.2e−16 for all groups, Chi-squared test). The high expression of TYRO3 in actively dividing cells was also observed in the TNBC-5, confirming the previous finding (*p*-value = 3.18e−8, Wilcoxon test, Supplementary Fig. 6l–m). We then investigated this TYRO3 expression pattern in another published cohort with 8 TNBC patient samples[30]. TYRO3 were lowly detected in 7 of the samples (detected in 1–8% of tumor cells). In the only sample

**Table 2 | Basic information for simulated datasets (scRNA-seq) for benchmarking experiments**

| Data | # Genes | # Cells | # Cell types | Sparsity |
|---|---|---|---|---|
| SD1-3 | 5000 | 4500 | 3; stable | ~65% |
| SD4-6 | 5000 | 4500 | 3; stable | ~78% |
| SD7-9 | 5000 | 4500 | 3; stable | ~90% |
| SD10-12 | 5000 | 4000 | 4; dynamic | ~65% |
| SD13-15 | 5000 | 4000 | 4; dynamic | ~78% |
| SD16-18 | 5000 | 4000 | 4; dynamic | ~90% |
| SD19-21 | 5000 | 4800 | 6; dynamic | ~65% |
| SD22-24 | 5000 | 4800 | 6; dynamic | ~78% |
| SD25-27 | 5000 | 4800 | 6; dynamic | ~90% |

(GSM4909284_TN-MH0114-T2) with relative high expression of TYRO3 (detected in 24% of tumor cells), the actively dividing cells showed higher expression of TYRO3 than other tumor cells (*p*-value = 0.039, Wilcoxon test, Supplementary Fig. 6n). These results indicated that the overall expression level of TYRO3 in breast cancer cells is highly patient specific, while the high-TYRO3 expressing samples always had TYRO3 preferably express in a small group of actively dividing cells. The GAS6-TYRO3 axis has been associated with tumor cell proliferation, malignancy, and anti-PD1/PD-L1 resistance in previous studies[31–35]. Thus, through the integration of data and prior knowledge using GSDensity, we postulated a potential role TYRO3 in TNBC proliferation using only a few TNBC samples with very sparse single-cell gene expression profiles and generated a testable hypothesis for further studies.

## Application of GSDensity in trajectory analysis to identify developmental stage related pathways

Current trajectory pathway analysis relies mostly on finding stage-related factors through factorization-based approaches[36,37] or finding stage-related gene co-expression modules[38], followed by manually annotating the factors and modules, while GSDensity could directly test curated gene sets and thus does not require efforts for factor

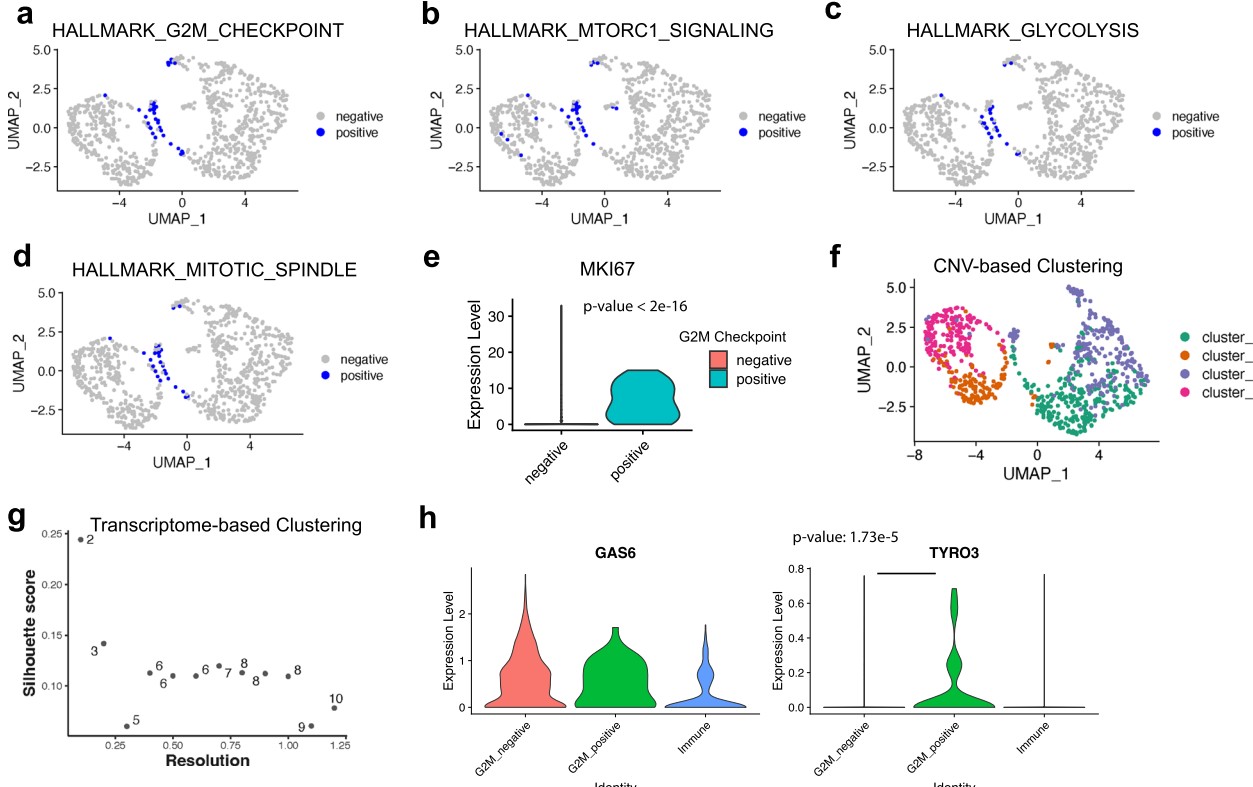

**Fig. 3 | GSDensity enables pathway centric analysis of tumor scRNA-seq data.** UMAP visualization of the TNBC-1 data using different hallmark gene sets (**a**: G2M checkpoint; **b**: mTORC1 signaling; **c**: glycolysis; **d**: mitotic spindle) to classify cells. Cells which are the most relevant to the hallmark are labeled as 'positive'. The UMAP was calculated using RNA expression (transcriptome) with default parameters using Seurat. **e** Violin plot visualization of the proliferation marker, MKI67, in TNBC-1 data, with the cells classified by G2M checkpoint gene set. Wilcoxon test (two-sided) was applied here. Source data are provided as a Source Data file for panels **e**–**h**. **f** UMAP visualization of the TNBC-1 scRNA-seq dataset. The cells are colored based on the clustering information from the inferred CNV profile. **g** The Silhouette scores and number of clusters out of choices of the 'resolution' parameter to cluster tumor cells in TNBC1 dataset with Seurat. The x-axis is the scanned 'resolution' (0.1 to 1.2) in Seurat 'FindClusters', and y-axis shows the Silhouette score. The text label (numbers) shows the number of clusters out of such parameter. **h** The expression level of GAS6 and TYRO3 genes in immune cells, G2M checkpoint positive, and G2M checkpoint negative tumor cells from the TNBC-1 data. Wilcoxon test (two-sided) was applied here.

identifications and annotation. To demonstrate this utility, we explored how GSDensity could synergize with pseudotime inference tools to find trajectory related pathways. We first performed pseudotime analysis on a subset of E17.5 mouse brain scRNA-seq data[39] (Fig. 4a, Supplementary Fig. 8a, b) and the direction of the trajectory was decided based on the expression of several developmentally related marker genes (Supplementary Fig. 8c). We applied GSDensity to this dataset and identified 97 KEGG and BIOCARTA pathway gene sets with significant coordination in some subpopulations of cells. We then grouped the cells based on pseudotime into equal sized partitions and calculated the pathway activity along the pseudotime trajectory by averaging single-cell PALs within each partition. We then clustered the fitted curves into 8 clusters using k-medoids clustering (Fig. 4b). Each of the 8 clusters except Cluster 6 had pathways specific to a different stage along the developmental trajectory, for example, Cluster 1 included the pathways enriched for the earliest stage while Cluster 2 for the following stage. Cluster 6 included pathways being relatively constant along the trajectory. We showed one pathway for each cluster as an example (Fig. 4c–j) and found several of them consistent with prior knowledge. For example, the term 'cell cycle' is the most highly relevant to the cells from the earliest developmental stage, since these cells are likely with the highest stemness and are actively dividing. Regulation of actin cytoskeleton also appeared to be an early pathway and it is known to be required for the cell migration during brain development[40]. Here we demonstrated the utility of GSDensity in identification of trajectory-related pathways.

## GSDensity identifies spatially relevant pathways

We first tested whether GSDensity could distinguish gene sets with spatial relevance from randomness by simulating ST datasets with four different modes: 'hotspot', 'hotspot with gradient', 'streak', 'gradient', inspired by a previous study[41] (Supplementary Fig. 9). The number and identity of spatially relevant genes were set through the simulation using SRTsim[42] (Methods). With PAL scores for pathways of interest, GSDensity can compare the background cell spatial density and the pathway-driven density by calculating the KL-Divergence between the two density distributions. It is worth mentioning that GSDensity was packaged in a flexible way that it can work with PAL scores from any PAL calculation methods for spatial relevance examination. For each of the simulated ST datasets, we visualized the spatial relevance of ground-truth gene sets and random gene sets, using a metric called 'delta-KLD' which equaled to $\log(D_{KL}(P||Q)) - mean(\log(D_{KL}(P_r||Q)))$, where $Q$ represents the background cell density, $P$ the pathway-driven cell density, and $P_r$ the cell density with shuffled cell-PAL scores. We calculated the PAL scores with GSDensity and CelliD, respectively, and observed that in both cases, GSDensity could distinguish ground-truth spatially relevant gene sets from random gene sets.

We applied GSDensity to a ST mouse forebrain dataset generated by the 10X Visium technology. We first clustered the data spots based on the transcriptome (Fig. 5a) and observed that all the clusters were also spatially segregated on the spatial map (Fig. 5b). Thus, the pathways with cluster-wise specificity would naturally display spatial relevance in this data. However, it is unclear whether there are high order

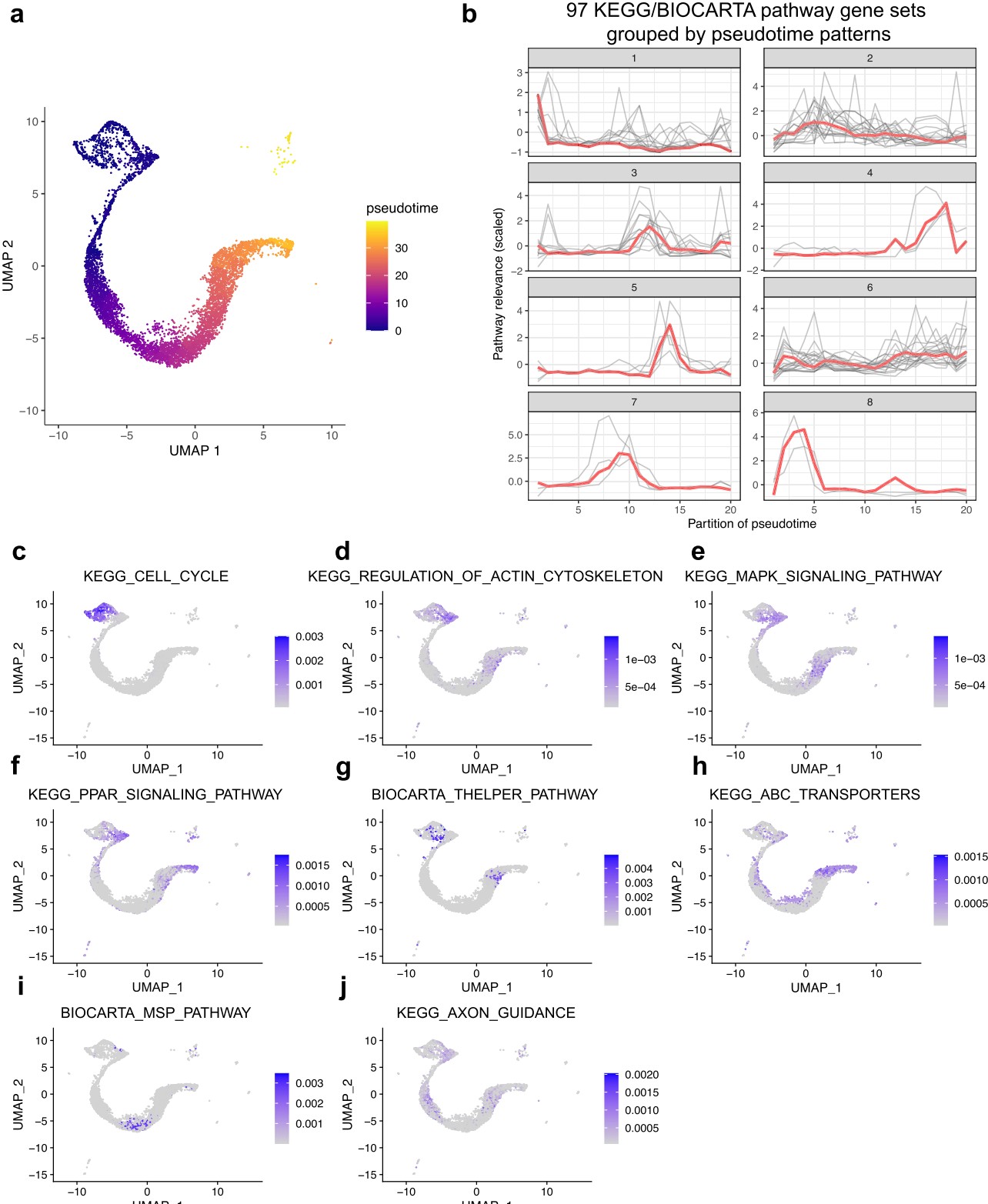

**Fig. 4 | Applying GSDensity to single-cell trajectory analysis reveals developmental stage related pathways. a** UMAP visualization of the inferred pseudotime in E17.5 mouse cerebral cortex data. **b** Pathway gene sets can be grouped into 8 clusters based on their pattern along the trajectory. Source data are provided as a Source Data file. Highlighting the cell relevance to a representative pathway from each of the 8 clusters (**c**: Cluster-1; **d**: Cluster-2; **e**: Cluster-3; **f**: Cluster-4; **g**: Cluster-5; **h**: Cluster-6; **i**: Cluster-7; **j**: Cluster-8).

organization of pathway activities across multiple clusters, which would be undetectable in cluster-centric analysis. To address this question, we first identified 727 GO biological process terms with coordination in the dataset using GSDensity. For each term, we calculated its spatial relevance and specificity for each cluster. The spatial relevance is quantified by KL-divergence between the pathway weighted kernel density estimation (KDE) and the equally weighted KDE (Methods). The specificity of a pathway for a cluster is quantified

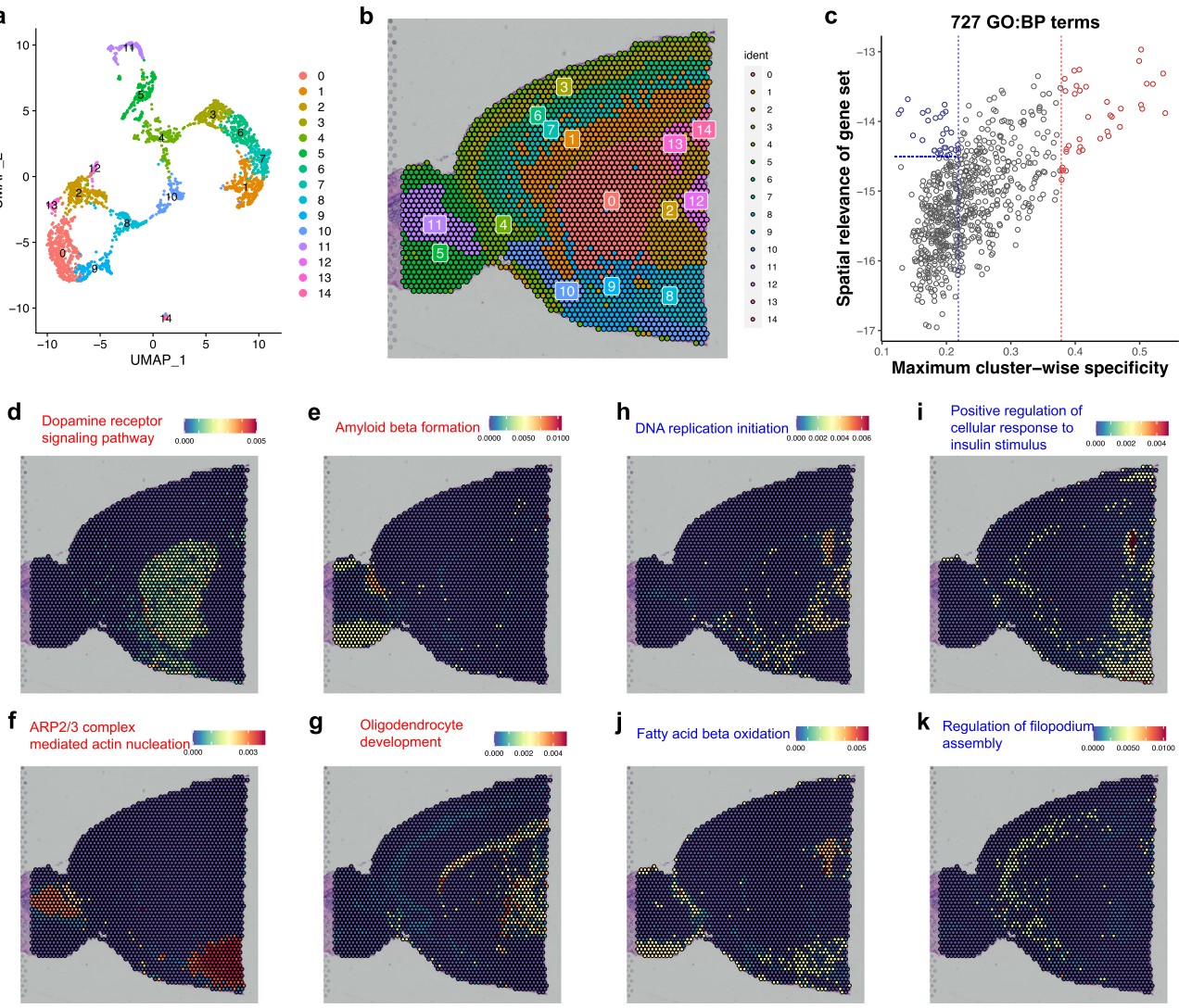

**Fig. 5 | Applying GSDensity to reveal spatial relevant molecular programs in mouse anterior brain. a** UMAP visualization of 15 cell clusters in the mouse anterior brain dataset. **b** Visualization of the 15 cell clusters in the spatial map. **c** Visualization of 727 gene ontology (biological process) terms based on their cluster-wise specificity and spatial relevance. The gene ontology terms are the ones having significant coordination in the mouse anterior brain dataset (Methods). The data points highlighted with red represent the ones with high cluster-wise specificity. The data points highlighted with blue represent the ones with low cluster-wise specificity and high spatial relevance. Source data are provided as a Source Data file. **d**–**g** Examples of gene sets with high cluster-wise specificity. **h**–**k** Examples of gene sets with low cluster-wise specificity but high spatial relevance.

by a specificity score based on Jensen-Shannon divergence, with larger values being more specific. We then plotted the 727 GO terms with their spatial relevance and maximum cluster-wise specificity (Fig. 5c). As expected, the spatial relevance showed largely positive correlation with cluster-wise specificity. Among the GO terms with high cluster-wise specificity (Fig. 5c, red) are dopamine receptor signaling for Cluster 1, amyloid beta formation for Cluster 5, ARP2/3 complex mediated actin nucleation for Cluster 8/11, and oligodendrocyte development for Cluster 2 (Fig. 5d–g). Interestingly, we also observed some GO terms with high spatial relevance and low cluster-wise specificity (Fig. 5c, blue). The cells highly relevant to these terms consisted of data spots from multiple clusters with higher-order spatial organization (Fig. 5h–k). For example, positive regulation of cellular response to insulin stimulus appeared highly active in the spots close to the caudal side (Fig. 5i), while fatty acid oxidation to the ventral side (Fig. 5j). It has been known that insulin receptors are expressed in hypothalamus and hippocampus[43] which are both located close to the caudal side of this anterior section. Although GSDensity was designed to perform cluster-independent data analysis, we demonstrated that

the pathway activity calculation by GSDensity can be easily integrated with cell information, such as cluster partition or spatial coordinates, when available.

Besides 10X Visium, multiple other ST technologies have been developed with whole transcriptome coverage. We applied GSDensity to a ST human prefrontal cortex dataset generated by the Slide-tags technology[44]. Cells were pre-annotated to several cell types as is shown in the UMAP and the spatial map (Supplementary Fig. 10a). Similarly, we identified 1061 GO terms enriched in some cell subpopulations, from over 7000 Biological Process terms using GSDensity (Supplementary Fig. 10b). Like the scenario of the mouse brain data, some pathways were both highly spatially relevant and highly cell-type specific, for example, protein localization to synapse was highly specific to inhibitory neurons and sensory perception of smell was highly specific to excitatory neurons, both of which occupied localizations with spatial relevance on the map (Supplementary Fig. 10c, d). On the other hand, we noticed some terms with relatively low cluster-wise specificity, and they turned out to be specific to subpopulations of cells. We found that among excitatory neurons, there were a subpopulation with

higher translational activity while another subpopulation with higher cell respiratory activity, and interestingly they appeared in two different layers (Supplementary Fig. 10e, f).

### GSDensity identifies common spatially relevant pathways in six cancer ST datasets

To investigate whether there were common spatially relevant pathways in different cancer types, we applied GSDensity to create a pan-cancer pathway activity map using six publicly available cancer ST datasets generated by the 10X Visium technology. The cancer types included were breast cancer (BC), cervical cancer (CC), intestinal cancer (IC), ovarian cancer (OC), prostate acinar cell carcinoma (PACC), and prostate cancer (PC). One section is analyzed for each cancer type. For each dataset, we tested the spatial relevance for over 10k pathway gene sets, including GO biological processing terms, MsigDB hallmarks, and canonical pathways from the MsigDB C2 collection. We identified 34 pathways that showed spatial relevance in all the six datasets (Supplementary Fig. 11 and Methods). To learn the relationship between these pathways and tumor cells, we first identified tumor cells based on their CNV profiles (Supplementary Fig. 12a–f), using 'normal' diploid cells as the control. We observed that the BIOCARTA granulocyte pathway displayed patterns that generally surrounded the tumor cells, which were visualized as contours (Fig. 6a–f, Supplementary Fig. 13). This observation was statistically significant (Methods, Chi-square test) in the PC ($p$-value = 1.575e−14), IC ($p$-value = 0.014), OC ($p$-value = 2.429e−12), and PACC ($p$-value = 0.030), while not for BC or CC. This observation suggested granulocytic infiltration towards the tumor cells, which has been found to be associated with tumor progression and metastasis[45–48]. We also saw that the spots with highly active granulocyte pathway are not homogeneously surrounding all the tumor clones, such as those in the cervical cancer (Fig. 6c) and in the prostate cancer samples (Fig. 6f). We also noticed that the mesenchyme morphogenesis pathway from GO biological process terms, showed spatial relevance in all the datasets (Supplementary Figs. 14 and 15). The data spots with high activity of this pathway overlapped with the tumor cells and appeared enriched at the borders of the tumor clones (Methods. for BC, $p$-value = 1.805e−11; CC, $p$-value < 2.2e−16; IC, $p$-value < 2.2e−16; PACC, $p$-value = 0.004). This is consistent with the current understanding that mesenchymal cells are highly related to tumor invasion[49,50]. Here we demonstrated that GSDensity could be used to infer tumor-TIME interface enriched pathways and identify recurrently activated pathways when multiple datasets were available.

## Discussion

Here, we report a computational framework, GSDensity, to perform directed pathway analysis on single-cell and ST data. Current practice in single-cell data analysis largely relies on assigning cells into discrete clusters and focuses on one or some clusters that appear interesting[6,51]. However, the clustering assignment process could be quite complex or even cause negative effect in many cases. For example, when the cells are sampled from a series of transition states or developmental states, it is impractical to group the cells into discrete clusters. Moreover, cluster-dependent analysis may suffer from the effect of 'double dipping', when the clustering is decided by gene expression profiles and the clustering is then used to learn differential genes or gene sets. On the other hand, in recent years we noticed several cluster-independent analysis tools for differential expression analysis[52,53] and differential cell enrichment analysis[54,55] showing superior sensitivity and resolution. GSDensity fills the gap that there are no dedicated tools for cluster-independent pathway analysis on single-cell data. For example, in this study, we showed that GSDensity can be used to directly classify cells using pathway relevance and perform downstream analysis based on the classification.

Through extensive benchmarking experiments using both real-world and simulated datasets, we showed that GSDensity performed consistently better than a set of popular frameworks in accurately score pathway activity on at single-cell level. The major challenge of this task is the sparsity and technical noise in scRNA-seq data. Embedding the data to lower dimensions is a useful method to enhance the signal-to-noise ratio. For example, factorization-based methods benefit from the lower-dimensional representation of data, however the interpretation of the 'factors' or 'patterns' is less straightforward. Moreover, the pattern discovery could have limited resolution due to both biological complexity and technical challenges. The application of MCA embedding in scRNA-seq studies was first reported in the CelliD[12] method, which generally performed the second best in the benchmarking. GSDensity uses network propagation to estimate the pathway relevance at single-cell level, while CelliD performs hypergeometric test on pathway gene sets and cell signature genes and reports transformed $p$-values. We consider that the pathway activity from network propagation is more smooth and easier to be implemented with other algorithms, such as what we did in treating the pathway activity as cell weights for spatial relevant pathway identification. Although varying by datasets, the cell embeddings and the gene embeddings produced by the MCA are largely comparable, as shown by the overlapping distance distributions in multiple real-world scRNA-seq datasets (Supplementary Fig. 16). In the MCA biplots, a cell-gene distance reflects the mutual specificity between a gene and a cell. The coordinates of the genes (or cells) are determined by the coordinates of the associated cells (or genes) weighted by their mutual specificity. The information of 'seed' genes can then be amplified by similar cells through highly specific cell-gene pairs. This design would make the size of neighborhood an important parameter when constructing the cell-gene graph. In our experiments, we found that the performance of GSDensity is robust and stable with respect to this parameter (Supplementary Figs. 17–19, Methods). With GSDensity, we also offer an option of PAL-based cell binarization for downstream analysis, as was demonstrated in the TNBC data analysis. Since the PAL calculation is an outcome of network propagation, for most coordinated gene sets (Fig. 1c), the PAL among all cells would have two or more modalities because the network propagation is restricted to the highly relevant cells and converges before propagating to less relevant cells. Thus, we recommend performing binarization on only gene sets that pass the coordination test. In our experience (e.g., PBMC data, GO Biological Process gene sets), over 98% of coordinated pathways (1160 gene sets from the GO Biological Process collection) demonstrated more than one modalities (adjusted $p$-value < 0.05, mode testing method by Ameijeiras-Alonso et al.[56]. using the R package 'multimode'). Occasionally, certain pathways can demonstrate more than two modalities. Thus, we recommend users to visually inspect the PAL distributions of important pathways to validate the automatic results and perform correction as necessary.

Given the rapid development of spatial genomics technologies[57,58], we also expanded GSDensity to identify spatially relevant pathways. Our approach benefits from the relatively accurate pathway activity scoring when the spatial information is not considered. Once the single-cell level pathway activity scores are computed, metrics or methods for spatial pattern discovery other than our weighted-KDE could also be applied, such as Moran's I for spatial autocorrelation. Also, as we pointed out, it is worth carefully considering the effect of cluster-specificity when we try to find spatially relevant pathways, given that the cluster information could display strong spatial relevance. In other words, the pathways showing both high cluster-specificity and high spatial relevance are likely the ones that can be discovered even without spatial information. GSDensity has two ready-to-use functions to compute the spatial relevance and cluster-wise specificity of pathways for dissecting these two situations.

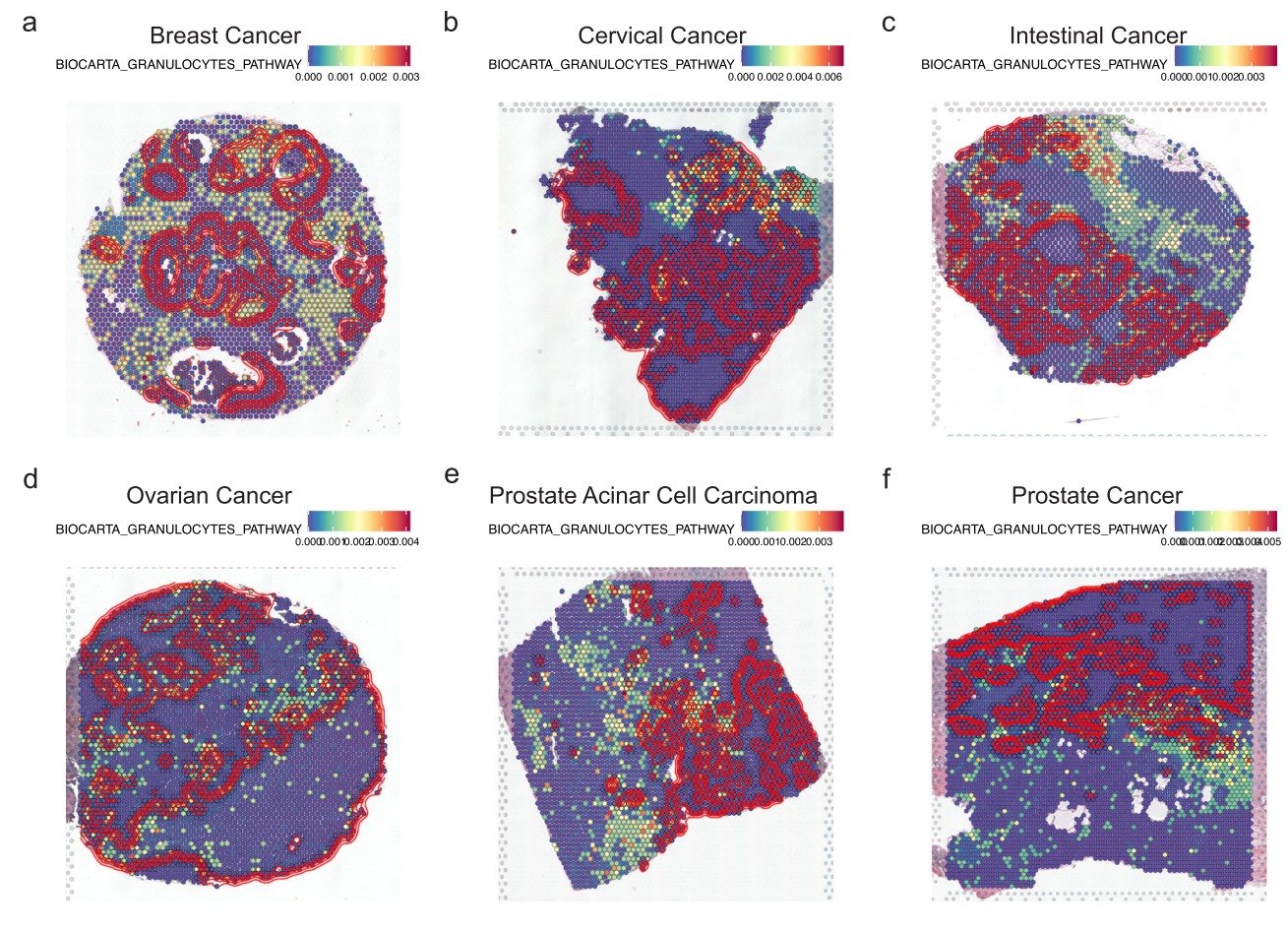

**Fig. 6 | Applying GSDensity to reveal the spatial distribution of granulocyte pathway in tumor samples of different tumor types.** Visualization of granulocyte pathway in breast cancer (**a**), cervical cancer (**b**), intestinal cancer (**c**), ovarian cancer (**d**), prostate acinar cell carcinoma (**e**), and prostate cancer (**f**). Red contour lines represent the density of tumor cells.

We demonstrated the application of GSDensity on scRNA-seq and ST data of the 10x Visium platform and Slide-tags, while GSDensity also has utility in other types of data. For example, GSDensity can do pathway analysis on Single-Nuclei Assay for Transposase-Accessible Chromatin with high-throughput sequencing (snATAC-seq) data, when gene expression can be inferred from the chromatin accessibility (often referred to as gene activity). GSDensity can also be applied to ST data generated by technologies such as seqFISH[59], seqScope[60], or DbiT-seq[60], since GSDensity requires only gene expression matrixes and spatial maps of cells as the inputs, not relying on any technique-specific data types. Also, GSDensity has a flexible framework that can test manually curated gene sets by users, such as the ones from CRISPR screening experiments, the ecotype features from Ecotyper[61], the multicellular programs discovered from DIALOGUE[62], or gene sets identified from large-scale datasets of molecular program discovery projects such as HTAN, HuBMAP, etc.

GSDensity can also be applied as a cell type annotator when marker genes were available. We showed that GSDensity performed best among pathway analysis tools in this task (Supplementary Fig. 4). For the benchmarking experiment, we used randomly truncated marker gene sets to control the uncertainty caused by variable gene set sizes. We further found that the performance of GSDensity in cell type prediction would increase when using full marker lists (with variable sizes) in most of the scenarios introduced in the 27 SERGIO-simulated datasets (Supplementary Fig. 20). We also compared the performance of GSDensity with another marker-based cell type annotation tool,

scSorter[63] and a clustering-based strategy for automatic cell annotation (Methods), using the 27 SERGIO-simulated datasets (Methods, Supplementary Fig. 21). We noticed that GSDensity and the clustering-based strategy generally out-performed scSorter. In the scenarios with dynamic cell states, especially as the data sparsity increases, GSDensity showed superior performance over the clustering-based strategy. It is worth noting that with an increase in cellular throughput, datasets with sparsity close to or higher than 90% became very common in recent single-cell studies[64]. Additionally, we examined automatic cell annotation in spatial transcriptomics data comparing GSDensity with DR.SC[65], a clustering-based strategy encouraging spatial-smoothness (Methods). With the labeled human tonsil data, we found that GSDensity performed better than the clustering-based strategy (59.32% vs 55.69%; Supplementary Fig. 22) indicating the importance of performing accurate molecular dissection in annotating ST data.

GSDensity is implemented in R with high efficiency. The largest memory cost was in the MCA embedding step, with about 25 Gb for a medium sized dataset ('hcabm40k' SeuratData dataset, 40,000 cells with 17369 genes). The runtime for the MCA step with different number of cells were tested (Supplementary Fig. 23a). It appeared increasing linearly with respect to the number of cells. The runtime for gene set coordination testing (without considering the MCA embedding calculation time) is more related to the number of gene sets instead of number of cells (Supplementary Fig. 23b). We further compared the pathway activity scoring speed among five methods: GSDensity, CelliD, AUCell, ssGSEA, and VAM (Supplementary Fig. 23c).

In general, CellID was the fastest, followed by GSDensity, AUCell, VAM, and ssGSEA. AUCell was faster than GSDensity when the number of cells was small while GSDensity was more scalable. GSDensity could calculate the pathway activity for 1000 gene sets in a dataset with 40,000 cells in about 10 min while AUCell took about 27 min. Options for parallel computing (through 'future.apply' R package) are implemented in GSDensity for gene set activity scoring, binarization, and spatial density evaluation and are highly recommended. Given the GSDensity algorithm, the tool relies on a background gene set to evaluate the density of pathway gene sets. Thus, GSDensity is applicable to spatial whole transcriptomic data but is not suitable for targeted gene data having limited numbers of genes. Imputation of unmeasured genes could be a strategy to overcome this limitation.

For integrated datasets from multiple origins with strong batch effects, since the MCA uses the original gene expression data, the batch effect correction should also be performed on the original gene expression data, instead of on the low-dimensional embeddings, using methods such as ComBat[66], CarDEC[67], or Scanorama[68]. We showed that although GSDensity performed reasonably well without having batch correction, using ComBat to adjust the gene expression data can improve its performance (Supplementary Fig. 24). Proper batch correction improves the performance of GSDensity, while over- or under-correction may introduce bias. Thus, we suggest to carefully evaluate batch correction outcome before performing pathway analysis. We also tested whether the performance of GSDensity will be affected by different normalization and transformation strategies for scRNA-seq data by comparing three strategies: the default Seurat normalization (also used by GSDensity by default), the SCTransform[69] implemented by Seurat, and the scran[70] strategy offered by the scran R package. We found that these normalization strategies gave highly consistent results for both gene set coordination test (Supplementary Fig. 25a–c) and gene set PAL calculation (Supplementary Fig. 25d).

Our original motivation of developing GSDensity is to build a tool for 'pathway-centric' analysis, with which we can dive into the data directly from the angle of pathways of interest. Initial gene sets could come from any prior knowledge, information, or preliminary data, such as RNA-seq data, CRISPR screen results, or GWAS genes, etc. When there lacks a prior knowledge regarding which specific pathways to test, users can start with carefully curated pathways in the public databases. For example, in the E17.5 mouse example, we used KEGG and BIOCARTA but not GO Biological Processes, because the latter included many cell type-related terms, while the mouse cells are largely un-differentiated at that stage. We chose to use GO Biological Processes for adult mouse brain ST data. For cancer ST data, since we aimed at finding common spatially relevant programs from multiple datasets, we chose to broaden our hypothesis space by using all gene sets from GO, Hallmarks, BIOCARTA, and KEGG.

We anticipate that GSDensity can play a critical role in meta-analysis type studies[71] focusing on specific pathways, for example, to find subgroups of tumor cells enriching the same pathway across many types of tumors. This could be very promising given the number of datasets collected nowadays. Additionally, many approaches in analyzing single-cell data can be transformed to 'assigning each cell a score using a gene set'. It is also possible to apply GSDensity in the identification of the mostly affected cells upon treatment of drugs that blocks the whole pathway; or finding the cells where a transcription factor (TF) is highly active given the targets; or discovering the subgroup of cells the most relevant to a trait when there is a set of GWAS genes available.

## Methods
### The MCA method for co-embedding cells and genes
The first step of GSDensity is to co-embed cells and genes into the same low-dimensional space using the MCA method. Here is a brief description of the calculation. Let $X$ be a scRNA-seq matrix with $K$ cells

$c_1, c_2, \ldots, c_K$ and $M$ genes $g_1, \ldots, g_M$ as rows and columns, respectively. A matrix $Y$ is then created as the binarization of each gene for each cell in the matrix $X$. For a cell $c_i$ in the matrix $Y$, the degrees of membership for gene $g_j$ are calculated by this membership function: $g_j^+ = \frac{X_{ij} - min(X_j)}{max(X_j) - min(X_j)}$ and $g_j^- = 1 - g_j^+$. Here $X_{ij}$ is the expression of $g_j$ in $c_i$ and $X_j$ is the vector of the expression of $g_j$ in all cells, so $Y$ has $K$ rows $(c_1, c_2, \ldots, c_K)$ and $2M$ columns $(g_1^+, \ldots, g_M^+, g_M^-)$. The '+' and '−' are seen as two categories for each gene and the values reflects the degrees of membership for each cell. This step is often referred to as 'fuzzy coding' to code a dataset with continuous measurement to categorical variables, required by the MCA algorithm[12,72]. The matrix of relative frequencies is $R = \frac{Y}{sum(Y)}$, and $sum(Y) = K \times M$. The matrix of standardized relative frequencies $Z$ is calculated by $Z = D_r^{-1/2} R D_c^{-1/2}$, where $D_r$ and $D_c$ are diagonal matrices with diagonal values equal to the row-sums ($R1$) and column-sums ($1^T R$) of $R$ ($1$ is a column vector of 1's). $Z$ is then decomposed $Z = UD_\alpha V^T$ where both $U$ and $V$ are orthonormal ($U^T U = I$; $V^T V = I$). $D_\alpha$ is the diagonal matrix with singular values. The coordinates of cells and genes are $C = D_r^{-1/2} U$ and $G = D_c^{-1/2} Z^T U$, relatively. Since this step is looking for a low-dimensional embedding, only the first several singular vectors are used and are tunable. Empirically setting 10 to 50 gives comparable results. This part was inspired by the previous study, CelliD, and we directly implemented its R function 'RunMCA' to perform this step in our method. Also as is implemented in CelliD, only the coordinates of $(\ldots, \ldots, g_M^+)$, representing the presence of the gene expression, are retained.

### Estimating and evaluating the density for gene sets of interest
To evaluate whether a gene set has true coordination, we need to estimate its density in the MCA space and compare that with the density of all genes. In this step, only the coordinates of genes, not cells, are considered, and the coordinates are scaled for each MCA dimension. To estimate the density of points in a space with more than 2 dimensions, GSDensity first select a few 'grid points' in the MCA space and estimate the local density of genes at these points as a representation of the overall density. This is inspired by a previous study, singleCellHaystack[52]. When selecting grid points, we partition the cells into $N$ groups of the same size, based on their distance in the MCA space, and thus the centroids are selected as grid points. This is to ensure that the grid points are relatively close to genes but not too close to each other. The grid point selection is done with the 'balanced_clustering' function in the 'anticlust' package[73]. The default number of $N$ is 100 and can be set by users. All the analysis in this manuscript used $N = 100$; a larger number of $N$ will allow for capturing more subtle alternations and on the other hand cost more computational time. For each grid point, the density is an aggregation of the density contributions of all data points. The density contribution $d_{ij}$ of a gene $g_i$ to a grid point $p_j$ is $d_{ij} = \exp(-\frac{dist(g_i, p_j)^2}{2})$. Here $dist(g_i, p_j) = \frac{Dist(g_i, p_j)}{bandwidth}$, where $Dist(g_i, p_j)$ calculates the Euclidean distances between two points. The $bandwidth$ is the median of the distance of all genes to their closest grid points.

With this method, we first calculate the density of all the grid points using all genes and these values are directly appended to form a background density distribution $Q$. $Q$ is a vector of length equal to $N$, the number of grid points, and is normalized that $sum(Q) = 1$. We then calculate the density of all the grid points using genes from specific pathways and form a density distribution $P$. $P$ has the same length as $Q$ and is also normalized to 1. The Kullback-Leibler divergence (KL-divergence) is calculated $D_{KL}(P||Q) = \sum_1^N P[x] \log(\frac{P[x]}{Q[x]})$. For each pathway, we create $L$ size-matched control gene sets by random sampling without replacement. We calculate their density $P_r^1, \ldots, P_r^L$ and

KL-divergence $D_{KL}(\mathbf{P}_r||\mathbf{Q})$ using the same method. By default, we set $L = 100$ as the number of control gene sets for each pathway tested. To perform statistical tests, we calculate $P(z \leq x) = \frac{1}{\sqrt{2\pi}} \int_{-\infty}^{x} e^{-t^2/2} dt$ where $x = \frac{\log(D_{KL}(\mathbf{P}||\mathbf{Q})) - mean(\log(D_{KL}(\mathbf{P}_r||\mathbf{Q})))}{sd(\log(D_{KL}(\mathbf{P}_r||\mathbf{Q})))}$ as the p-value. We perform this using the 'pnorm' function in R, for each pathway considered. Quantile-quantile plots showed that the $\log(D_{KL}(P_r||Q))$ values approximated normal distributions for random gene sets of different length, from 30 to 500 (Supplementary Fig. 26). When multiple pathways are tested at the same time, the p-values will be FDR-adjusted.

## Calculating the gene set activity at single cell level

Network propagation has been widely applied to the analysis of gene-gene networks for candidate gene prioritization[74] and recently cell-cell networks to find phenotype relevant cells[75]. Here network propagation is applied to the calculation of gene set activity scores for each cell. We first construct a nearest neighbor graph with cells and genes as nodes based on their Euclidean distance in the MCA space. The default number of neighbors is set to be 300 and can be tuned by users. We have evaluated the robustness of GSDensity performance to this parameter in terms of pathway activity in the PBMC data (Supplementary Fig 17) and in terms of cell type classification in the 8 real-world datasets (Supplementary Figs 18 and 19). We found that, consistently across datasets, the performance of GSDensity increases as the number of neighbors increases from 20 to 200 and reaches plateau afterwards. This graph is unweighted and undirected and is converted to a simple graph with no multiple edges. We then use random walk with restart (RWR) for the label propagation, where the genes in the pathway of interest are used as 'seeds'. The restart rate is set to 0.75 by default, and the convergence will be reached when the L1-norm between the state matrices of two consecutive steps is less than 1e-6. The RWR step is done with calling the 'dRWR' function from the 'dnet' R package[76]. The output of the RWR is a vector $\mathbf{H}$ with length equal to $M + K$, and each item represents the relevance between the cell or gene to the pathway. We subset $\mathbf{W}$, a vector of length $K$ from $\mathbf{H}$ which only contains items as cells and normalize $\mathbf{W}$ that $sum(\mathbf{W}) = 1$. This vector $W$ is then the gene set activity scores for the cells.

## Using weighted kernel density estimation to find spatially relevant pathways

To test whether a pathway is spatially relevant, we require a pre-compute of the activity score of cells for that pathway ($\mathbf{W}$), and the two-dimensional spatial coordinates for the cells, $\mathbf{F}$. $\mathbf{F}$ is a $K \times 2$ matrix with rows for cells and columns for coordinates (the x-y plane). Grid points are uniformly picked along the two axes. The density of a grid point $p_j$ is $d_j = \frac{\sum_i^K w_i \varnothing((x_{p_j} - x_i)/h_1) \varnothing((y_{p_j} - y_i)/h_2)}{K h_1 h_2}$, where $w_i$ is the weight for cell $c_i$, $x_i$ and $y_i$ are coordinates of $c_i$, $K$ is the number of cells, $h_1$ and $h_2$ are estimated bandwidths for the x and y axes, and $x_{p_j}$ and $y_{p_j}$ are coordinates of $p_j$. The bandwidths $h_1$ and $h_2$ are decided using the 'bandwidth.nrd' function in the 'MASS' R package. $\varnothing$ calculates the probability density of standard Gaussian distribution: $\phi(x) = \frac{1}{\sqrt{2\pi}} e^{-x^2/2}$. This function is different from the 'kde2d' function in the 'MASS' R package only by the $w_i$ item. The implementation of this algorithm is from the online resource: https://stat.ethz.ch/pipermail/r-help/2006-June/107405.html. We first compute a background cell density by setting all $w_i$ equal to $\frac{1}{K}$. For each pathway, we calculate a pathway-specific cell density by directly using items in $\mathbf{W}$ (pathway scores) as the corresponding $w_i$. In both the background cell density and the pathway-specific cell density, the weights aggregates to 1, and the output density distributions are thus directly comparable. Here, comparing the two density distributions is performed the same way as we described in 'Estimating and evaluating the density for gene sets of interest'. We also used randomization to evaluate the statistical significance of the pathway-specific cell density.

## Curation of real-world datasets

We collected 8 real-world datasets for validating the performance of GSDensity and compare it with several other tools. The selection of real-world datasets was generally based on the availability of pre-annotation (by original data generators) labels and cell type markers in public databases, to ensure the fairness of benchmarking. We also tried to cover different types of tissues or organs. General information for these datasets can be found in Table 1. Markers for cell types were searched from three public databases: XCell[15], PanglaoDB[16], and CellMarker2[17]. The marker curation was performed in a sequential way by searching XCell first, then PanglaoDB then CellMarker2 until a hit is found. The cell type annotation was matched with the cell type names in these databases to curate the marker list for the cell type. The cell types were removed when markers cannot be found in any of the databases. The code for curation the datasets and marker lists, as well as the marker collection files, were available at https://github.com/qingnanl/GSDensity_manuscript_code. For the 'heart' dataset, the R object needs to be downloaded from https://www.ncbi.nlm.nih.gov/geo/query/acc.cgi?acc=GSE183852. For the 'lung' dataset, the expression matrix needs to be downloaded from https://singlecell.broadinstitute.org/single_cell/study/SCP886/hca-lungmap-covid-19-pittsburgh-lafyatis-2019-morse#study-download. The other datasets were curated by either 'SeuratData' or 'scRNA-seq', both being R packages.

## Simulation of scRNA-seq datasets with SERGIO

SERGIO required two inputs for generating steady-state datasets: the expression of master regulators for each cell type and the relation between each gene and its regulator(s). In SERGIO, genes could be defined as master regulators (cannot be target; must predefine), regulators (can regulate other genes and be regulated by other regulators or master regulators), and targets (can only be regulated by regulators or master regulators). To ensure the reliability of ground truth for the purpose of this study, we did not include regulators in the simulation. We simulated 3 steady-state datasets from Mode-1 (3 types, 1500 cells per type, 5000 genes) by randomly generating master regulator expression matrices (from uniform distribution from 0.1 to 5) and randomly defining the relationship between master regulators and targets three times. In each time, we defined 25 genes as master regulators (0.5% of all genes), 500 genes as strong targets (10%), 500 as medium targets, 500 as weak targets. Each of these 1500 target genes had one master regulator attributed through random sampling. The differences among the strong, medium, and weak targets are the 'strength' parameters in SERGIO. This was a hyperparameter related to the contribution of the regulator towards the target. Strong targets had the absolute value of the strength parameters ranging from 2 to 5, with medium ones having 1 to 2 and weak ones having 0.5 to 1. The strength parameter could be positive or negative with 0.75 and 0.25 probabilities, respectively, defined by the author, to better represent real single-cell datasets. Negative targets were not used as markers for the following benchmarking experiments. The rest of the genes were assigned multiple master regulators.

With these inputs and other default SERGIO parameters, 'clean' single-cell gene expression matrices could be simulated and could be modified to add noises afterwards. We demonstrated the UMAP of one example with cell types labeled using the clean matrix of Mode-1 in Supplementary Fig. 27a. The overall expression level of the strong, medium, and weak markers for each cell type were verified with the 'AddModuleScore' function of Seurat, shown in Supplementary Fig. 27b. Multiple types of noises could be simulated by SERGIO, and to ensure simplicity, we fixed most of the noise parameters and only tuned the 'percentile' parameter for dropout simulation, to make the final matrix with dropout rate at roughly three levels: 65%, 78%, and 90%. Thus, for each of the 'clean' matrix, three sparse matrices were generated. We showed the UMAP of one example (65% drop-out rate)

in Supplementary Fig. 27c and the verification of marker sets with different specificity levels in each cell type (Supplementary Fig. 27d).

For simulating 'dynamic' datasets to represent continuous or developmental data, an extra input, in the format of a cell-type by cell-type matrix, is needed to specify the migration rate from one cell type to another. We used a bifurcation model and a trifurcation model previously reported by the SERGIO package. For the bifurcation model (Mode-2, Supplementary Fig. 28a), we simulated 3 datasets (4 cell types, 1000 cells per type, 5000 genes). For the trifurcation model (Mode-3, Supplementary Fig. 29a), we simulated 3 datasets as well (6 cell types, 800 cells per type, 5000 genes). For both models, we defined 50 genes as master regulators (0.5% of all genes), 500 genes as strong targets (10%), 500 as medium targets, and 500 as weak targets, similar to the case of steady-state simulation. In these cases, the expression of the rest of the genes were generated randomly and combined with the master regulator and target expression to ensure reasonable runtime and RAM usage. Quantile normalization was performed before each combining. It is also worth mentioning that SERGIO actually generated both a 'spliced' and an 'un-spliced' count matrices, and here we only used the spliced one. Similar to the clean matrices of Mode-1, we simulated the noise to reach final matrices with roughly 65%, 78%, and 90% drop-out rates. We demonstrated the UMAP of clean matrices of Mode-2 and Mode-3 and the overall expression of the marker sets, as well as the matrices with noise added and the overall expression of the marker sets in Supplementary Figs. 28b–e and 29b–e.

All the code used for generating the SERGIO inputs, running SERGIO, adding noises, and the dynamic migration matrices, were available at: https://github.com/qingnanl/GSDensity_manuscript_code.

### Ground-truth marker gene sets in simulated datasets

The SERGIO simulation used pre-defined master regulator expression as a basis. In our practice, the master regulator expression is an outcome of random sampling from a uniform distribution between 0.1 and 5. For each cell type, we quantify the specificity of the master regulator by

$$Specificity\ score = \frac{Expression\ of\ the\ master\ regulator\ in\ cell\ type\ A}{\max(Expression\ of\ the\ master\ regulator\ in\ other\ cell\ types)}.$$

Top three master regulators with highest specificity scores for each cell type were defined as cell-type specific regulators, and their targets with positive regulatory strength were then defined as ground-truth marker genes.

### Benchmarking the performance of GSDensity

For the analysis testing whether GSDensity can distinguish gene sets with true coordination (Fig. 2b–e, Supplementary Fig. 2), we created four groups of gene sets based on marker genes and random sampling. For the all-marker sets, we generate 5 sets for each marker gene list by downsampling to 80% of its original size without replacement (notice that the curated marker gene sets have different sizes for different cell types). For the markers with size-matched random genes, the 'marker.mix.1' and 'marker.mix.3' sets, we downsampled each marker gene sets to 40% size and 20% size and mix them with 1:1 randomly sampled gene and 1:3 randomly sampled genes, respectively. This ratio is designed to ensure these two types of synthesized gene sets have the same size of the all-marker sets. Again, for each marker gene sets, we generated 5 sets for each of the two categories. Lastly, we generated random gene sets of matched sizes with the marker sets and synthesized gene sets. We applied GSDensity with these gene sets to evaluate the statistical significance of their coordination. One example (Fig. 2a) was used to show the density of gene sets intuitively. Only for the visualization purpose, the MCA coordinates of cells and genes were first embedded to two dimensions using UMAP and the density was plotted according to the UMAP coordinates.

To compare GSDensity with current methods in the aspect of single-cell pathway activity calculation, we designed two

benchmarking strategies using pre-annotated scRNA-seq data and curated gene sets, to ensure unbiasedness (Supplementary Fig. 1c, d).

For the AUC metric (Supplementary Fig. 1c), we assess the capability of recovering the correct cell type with PAL scores from each method. For constructing the recovery curves, we order the cells based on their PAL score, and calculate the ratio of recovery at sixteen points: 0.025, 0.05, 0.075, 0.1, 0.125, 0.15, 0.175, 0.2, 0.3, 0.4, 0.5, 0.6, 0.7, 0.8, 0.9, 1.0. At each point, we recover the cells with top PAL scores (e.g., at 0.025, we recover the top 2.5% of cells with highest scores) and calculate the proportion of the recovered "correct cells" in all the "correct cells", which thus gives a value for the y-dimension of the recovery curve. The AUC score of the curve is calculated using the function 'AUC' of the package 'DescTools'. For real-world datasets, besides the original marker gene sets, we generated gene sets of five other categories, by reducing the size of marker gene sets and mixing them with randomly sampled genes. This was because in real data analysis, pathway genes were not as specific as marker genes in most cases, and it was useful to benchmark the performance of tools when the pathways were not highly specific. Unlike simulated datasets, we did not have prior knowledge on the specificity level of each marker gene to the cell type. When generating downsampled marker gene sets or mixture gene sets, we performed random sampling five times for each of the marker gene sets, so there were more data points for downsampled marker gene sets or mixture gene sets. For simulated datasets, we applied the AUC metric using marker sets with different specificity levels.

For the ACC (accuracy) metric (Supplementary Fig. 1d), we used PAL scores in a competitive way for cell type prediction. For each cell in a dataset, we compute PAL scores for all the marker sets and assign the cell's identity to the set achieving the highest PAL score. The performance of each method could then be assessed by the accuracy of such prediction. To control for difference in variable gene set sizes, we randomly downsampled all the gene sets to 10 genes before calculating the PAL and repeated each experiment five times.

We compared GSDensity with AUCell[11], CelliD[12], GSVA[19], ssGSEA[10], scGSEA[21], and VAM[20]. We used the default parameter for all the six methods. We used the 'gsva' R package to perform GSVA and ssGSEA. For benchmarking the runtime of GSDensity and other methods, we used the public dataset 'hcabm40k' (SeuratData R package) which has 40,000 cells. We randomly selected gene sets from the Gene Ontology Biological Processes database, as input to the methods. We dissected GSDensity into three parts for runtime benchmarking: calculating MCA embeddings (Supplementary Fig. 23a), testing gene set coordination (Supplementary Fig. 23b), and calculating pathway activities (Supplementary Fig. 23c). We compared the pathway activity calculation speed between GSDensity and several other methods. For all the methods, preprocessing steps were not recorded in the runtime benchmarking, such as the MCA calculation for GSDensity and CelliD, the pre-calculation of gene rankings for AUCell, etc. The runtime recording was performed using the R package 'microbenchmark'. The runtime benchmarking was performed using Linux (Redhat Enterprise Linux) system, with 12 cores, 120 G RAM.

### General single-cell data analysis

We used Seurat 4.0[77] for most of the data wrangling and visualization, for both scRNA-seq and ST data. The pathways used for this study were all from the MSigDB database[26,78,79], using the 'msigdbr' R package. Default parameters were used for GSDensity analysis. ComBat from R package 'sva' was used for batch correction (Supplementary Fig. 24).

### The TNBC data analysis

The TNBC data was downloaded from GSE148673 and GSE161529 in the format of gene expression matrices[25]. The 'TNBC-1' data (from GSE148673) was mainly used for discovery and others were used for possible validation purpose. Tumor cells from the TNBC-1 data was

identified using the 'copyKat' R package[25] with default parameters and the number of clusters were set to 4. GSDensity was applied only to the tumor cells to score them based on their relevance to a set of hallmark gene sets. To binarize the cells, GSDensity called the 'loc-modes' function of the 'multimode' R package[80] to identify the anti-mode of gene set activity of cells and used the antimode for the binarization.

Fibroblast cells (normal) and immune cells were identified using previously reported markers. The immune cells were then used to infer the ligand-receptor based cell-cell communication with classified tumor cells using the 'cellchat' R package[29]. The 'negative' cells for G2M checkpoint were downsampled to the size of the 'positive' cells to avoid that the comparison between groups being driven by the number of cells.

**The E17.5 mouse brain data analysis**
The data was downloaded from GSE153162 in the format of an '.h5' file[39]. We performed standard clustering and removed the discrete clusters from the data to ensure the reliability of the trajectory inference. The trajectory inference was performed using 'monocle3' and Cluster 6 was set to be the root, given its specific expression of *SOX2*. We further verified the trajectory by exploring the expression of several known genes specific for different developmental stages (Supplementary Fig. 8c). GSDensity was used to identify KEGG and BIOCARTA pathways with true coordination. The reason we only used these two pathway curations was that the cells were largely in developmental stages, and we did not want to use pathway curations with many cell-type specific terms (e.g., GO terms) due to lacking interpretability in this case. From all the input pathways, we identified 97 pathways with coordination. We equally split the cells into 20 partitions along the pseudotime trajectory based on the inferred pseudotime for the purpose of quantifying pathway activity along trajectory. These partitions were not related to the original clustering shown in Supplementary Fig. 8a, b. We then averaged the pathway scores within each partition, so each pathway was summarized into a curve with 20 ordered data points. We then applied k-medoids clustering for these pathway-time curves using the 'pam' function of the 'cluster' R package. We plotted one pathway as an example for each of the clusters (Fig. 4c–j).

**Simulation of ST data with SRTsim**
We simulated ST datasets using SRTsim[42] with four modes: 'hotspot', 'hotspot with gradient', 'streak', 'gradient'. We used the reference-free mode to generate these datasets with R-Shiny. Briefly, cell positions were randomly initiated, and clusters were created using the lasso tool in the R-Shiny application of SRTsim. The expression levels of ground-truth signal genes were designed to increase as new cluster is generated (in an alphabetical order, for example, Cluster B had higher expression levels on the signal genes, compared with Cluster A). Parameters used for simulating the datasets were listed in Table 3.

**The mouse brain and human prefrontal cortex ST data analysis**
The mouse brain ST data was obtained from the 'stxBrain' dataset of the 'SeuratData' R package using only the 'anterior1' subset. The data

preprocessing and clustering was completely following the online tutorial (https://satijalab.org/seurat/articles/spatial_vignette.html).

We applied GSDensity analysis for this data treating each data spot as a single cell and used all the GO biological process terms to test their coordination. We then plotted the spatial relevance and cluster-wise specificity only for the gene sets with significant coordination. The highlighted gene sets are either with maximum cluster-wise specificity larger than its 95% quantile (red) or with maximum cluster-wise specificity smaller than its 50% quantile and spatial relevance higher than 80% of its quantile (blue).

The cluster-wise specificity (CWS) between a cluster $L$ and a pathway $A$ is $CWS(L,A) = 1 - \sqrt{JSD(\mathbf{I}_L, \mathbf{W}_A)}$. $JSD$ is to calculate the Jensen-Shannon divergence between two vectors. $\mathbf{I}_L$ is a vector $i_L^1, i_L^2, \ldots, i_L^K$ of length $K$ (number of cells) initiated as $i_A^j = \begin{cases} 1, & \text{if cell } c_j \in \text{ cluster } L \\ 0, & \text{if cell } c_j \notin \text{ cluster } L \end{cases}$ followed by normalization to let $sum(\mathbf{I}_L) = 1$. $\mathbf{W}_A$ is a vector that contains the single-cell level gene set activity scores for pathway $A$. This way of defining the cluster-wise specificity was previously reported[81,82]. The CWS computation was packaged with a ready-to-use function of the GSDensity package.

The human prefrontal cortex dataset was downloaded from https://singlecell.broadinstitute.org/single_cell/study/SCP2167/slide-tags-snrna-seq-on-human-prefrontal-cortex#study-download. The cell type annotation was included in "humancortex_metadata.csv", and the spatial coordinates were included in "humancortex_spatial.csv". We used Seurat for preprocessing of the data. Other parts of the analysis were the same as described in the analysis of the mouse brain ST dataset.

**The cancer ST data analysis**
The cancer ST datasets were downloaded from the 10x genomics website, and all the six datasets were collected with Visium for FFPE and preprocessed with Space Ranger 1.3.0. Clustering and preprocessing of the datasets were following the same tutorial for the mouse brain ST data analysis. We applied GSDensity analysis for this data treating each data spot as a single cell and used over 10k gene sets including GO biological processing terms, MSigDB hallmarks, and canonical pathways from the MSigDB C2 collection to test their coordination. We first tested the pathway coordination only considering the gene expression and then used the pathways with coordination to test their spatial relevance. FDR-based multi-testing correction was applied to the results of both testes. Thus, for each dataset, we ended up with a list of pathways that are heterogenous and spatially relevant. To visualize the pathway activity and tumor cells, we first used the R package 'copyKat' to predict the identity of tumor cells using transcriptome information. Immune cell clusters were first identified using several markers (CD4, CD3E, PTPRC, NKG7, CD3D, CD14) and were used as normal cell controls for the copyKat analysis. The tumor cells were then visualized with their two-dimensional density. To perform enrichment analysis of certain pathways in tumor boundary cells, we partitioned the cells in the ST datasets in two ways. First, we partitioned the cells based on their relevance to the pathways of interest (granulocyte pathway and mesenchyme morphogenesis pathway in our cases) the same ways as demonstrated in the TNBC analysis (Methods). Second, we partitioned

**Table 3 | Basic information for simulated datasets (ST) for benchmarking experiments**

| Name | Cells | Genes | Sparsity | Dispersion | Mean |
|------|-------|-------|----------|------------|------|
| Spot | 4059 | 500/500/5000 | 0.85 | 1 | 1 |
| Spot.grad | 4457 | 500/500/5000 | 0.85 | 1 | 1 |
| Streak | 4457 | 500/500/5000 | 0.85 | 1 | 1 |
| Grad | 4657 | 500/500/5000 | 0.85 | 1 | 1 |

The column of genes showed the number of positive signal genes, negative signal genes, and random noise genes. The sparsity, dispersion, and mean were input parameters for SRTsim.

the cells based on whether they were boundary cells, which were defined as those with at least 1 and at most 4 of the neighbors being tumor cells, considering the tightly packing pattern of Visium data. We then performed Chi-square test to examine whether boundary cells enrich cells of the relevant pathways.

## Marker-based cell type annotation

We applied scSorter[63] on the 27 SERGIO-simulated datasets, using the same marker sets that we used for benchmarking GSDensity with other pathway scoring tools. Default settings were used as described here: https://cran.r-project.org/web/packages/scSorter/vignettes/scSorter.html. We also employed a 'cluster plus marker' strategy (Supplementary Fig. 21) for cell type annotation. For each dataset, we first used Louvain clustering (resolution 1.0) to cluster cells. We then use the "AddModuleScore" function to calculate the normalized average expression of each marker set in single cells and calculate the average marker expression level for each cluster. After scaling, we annotate the cluster based on the highest average marker expression level. The ACC metric was the same as described above to evaluate the method performance. For the 'cluster plus marker' strategy with spatial genomics data, we used DR-SC[65] instead of Louvain clustering.

## Statistics and reproducibility

No statistical method was used to predetermine the sample size. We ensure replicates when random sampling was involved in simulations, ranging from 3 (generation of simulation data of the same condition) to 40 (simulating coordinated and random gene sets) replicates. No data were excluded from the analyses. The experiments were not randomized. The Investigators were not blinded to allocation during experiments and outcome assessment. Student's $t$ test (one-sided) was used for gene set coordination test and Wilcoxon test (two-sided) was used to for differential gene expression examination. For all boxplots used in this manuscript, the center line of the box plot showed the median of data; the box limits showed the upper and lower quartiles; the whiskers showed 1.5 times interquartile range. When individual data points were not displayed, points show outliers.

## Reporting summary

Further information on research design is available in the Nature Portfolio Reporting Summary linked to this article.

## Data availability

All relevant data supporting the key findings of this study are available within the article and its Supplementary Information files. All real-world datasets used in this study were downloaded from public resources. Source data are provided with this paper. The PBMC, BMCITE, pancreas (for batch correction), hcabm40 (for runtime benchmarking) scRNA-seq datasets and the mouse brain ST dataset were obtained from the SeuratData R package. The human cortex, pancreas (for benchmarking), spleen immune, and liver immune scRNA-seq datasets were obtained from the 'scRNAseq' R package [https://bioconductor.org/packages/3.16/data/experiment/html/scRNAseq.html]. The heart scRNA-seq data was downloaded from GEO with accession number "GSE183852". The lung scRNA-seq data was downloaded from the Broad Institute single-cell portal with accession number "SCP886". The TNBC data was downloaded from GEO with accession numbers "GSE148673" and "GSE161529". The developmental mouse brain data was downloaded from GEO with accession number "GSE153162". The Slide-tags data was downloaded from the Broad Institute single-cell portal with accession number "SCP2167" and "SCP2169". The tumor ST data was obtained from 10x genomics [https://www.10xgenomics.com/resources/datasets?menu%5Bproducts.name%5D=Spatial%20Gene%20Expression&query=&page=1&configure%5Bfacets%5D%5B0%5D=chemistryVersionAndThroughput&configure%5Bfacets%5D%5B1%5D=pipeline.version&configure%5BhitsPerPage%5D=500&configure%5BmaxValuesPerFacet%5D=1000].

## Code availability

The GSDensity software is available at GitHub: https://github.com/qingnanl/gsdensity. Analysis code[83] is available at GitHub: https://github.com/qingnanl/GSDensity_manuscript_code.

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

## Acknowledgements

This work was supported by NIH grant U01CA247760 to K.C. and Human Cell Atlas Seed Network Grant CZF2019-02425, CZF2019-002432, and CZF 2021-239847 to K.C. from the Chan Zuckerberg Initiative. We thank Drs.Jinzhuang Dou, Shaoheng Liang, Vakul Mohanty, and Merve Dede for their valuable suggestions.

## Author contributions

Q.L. and K.C. conceived the study. Q.L developed the GSDensity method, implemented the software, analyzed data, and prepared figures. Y.H and S.H tested the package in different environments and datasets. Q.L, Y.H, and S.H debugged and optimized the package. Q.L and K.C. wrote the manuscript with input from all authors. The authors reviewed, edited, and approved the manuscript. K.C. supervised the project.

## Competing interests

The authors declare no competing interests.
