## [Peer Review File · Nature Communications]

Pathway Centric Analysis for single-cell RNA-seq and Spatial Transcriptomics Data with GSDensityReviewer #1 (Remarks to the Author):

The authors presented GSDensity, which is a pathway-centric analysis method for scRNA-seq and spatial transcriptomics data. Using the MCA method to project cells and genes to a common latent space, GSDensity finds cells that are associated with a given gene set. The manuscript includes results found by this method which are potentially biologically meaningful. Overall, the idea of using gene sets to find relevant cell sets is interesting and a method with this purpose can be a valuable addition to the current single cell methods. The manuscript, however, should be improved:

1. From the description of the methods and the demonstrated use cases, GSDensity takes a given gene set, evaluates whether it has true coordination, and finds cells for which these genes are major contributors of their profiles and possibly functions. It appears that users need to pre-define a set of candidate gene sets, and evaluate and analyze them one by one. GSDensity does not provide gene sets. The abstract should make this point clear. The current wording like "detect biologically distinct gene sets" and "identify pathways that are active at various stages" could also mean that GSDensity finds such gene sets of pathways from data.
2. The manuscript includes results from different datasets and cell types, and different initial gene sets or pathways were used as input to GSDensity to test. It would be helpful if the authors can provide guidance on how to obtain the initial sets of pathways to test, and ideally provide a generic set of pathways for users to use as default.
3. Some terms used in the manuscript need further clarification or definition. For example, in the Introduction, it's not clear what "heterogeneity of a pathway" means. Also, the "coordination" of a gene set is not a term that is commonly used.
4. MCA constitutes the main part of the method of GSDensity, and the success of GSDensity in identifying cells most relevant to a set of genes relies on that the distance between cell embedding and gene embedding is meaningful, in the sense that the cell embedding and gene embedding are comparable. In addition to giving the MCA calculation process which is included in Methods, the authors are suggested to discuss and justify that the cell embedding and gene embedding are comparable.
5. GSDensity constructs a nearest neighbor graph as a key step in its pipeline. The size of the neighborhood is an important parameter. It's important to show if the results are sensitive to this parameter and how should users set this parameter in practice.
6. In Fig. 4b, it seems that 20 partitions are used. How are these partitions obtained? Are these the clusters shown in Extended Data Fig. 8b? It is confusing because Data Fig. 8b has 12 clusters instead of 20.

Minor points:

- In Fig. 2, please use uppercase and lowercase letters properly for the methods' names.
- Fig. 3i was referred to in the manuscript but the plot was not labeled separately.
- Line 97 should refer to the Methods section here.

Reviewer #2 (Remarks to the Author):

In this manuscript, the authors proposed GSDensity, a computational method that uses multiple correspondence analysis (MCA) to reconstruct pathway activity at the single-cell level. Although pathway analysis using single-cell datasets is not a brand-new topic, the author compared their method with other state-of-the-art methods on several simulated and real datasets. They show that their method usually performs similarly or better than CellID, another popular method to perform pathway analysis on single-cell data and based on multiple correspondence analysis like GSDensity. Although interesting, the manuscript is more challenging to follow in some parts (particularly the paragraph on TNBC) that should be rewritten to be more straightforward. Here

are some comments that I believe could help in improving the manuscript:

1. Scalability is a crucial issue when it comes to single-cell analysis. However, no details about computing time are presented in the article. Can the authors deepen this aspect and compare their method with the others in terms of average running time at increasing number of considered cells (i.e. from 10,000 to 50,000 or more) and pathways? This analysis could also help the authors show an additional advantage of their method compared to cellID, which performs similarly on many occasions.
2. In GSDensity, the authors apply MCA to data normalized with Seurat. Can the normalization affect the results of the enrichment analysis? Can the authors try to decompose the data matrix resulting from a different normalization procedure (i.e. $\log(\text{CPM}+1)$ or scanpy normalization and so on)?
3. Another important noisy aspect of single-cell data is the batch effect. Can the others evaluate how batch affects pathway activity? For this, the authors can use many available datasets in the literature. See, for example, <https://www.nature.com/articles/s41592-019-0619-0> or <https://www.nature.com/articles/s41592-018-0254-1>
4. It Would be nice to add comparison with at least another method based on low dimensionality spaces like the recently published scGSEA you can find here <https://academic.oup.com/nargab/article/5/1/lqad024/7069282>
5. Paragraph "Classification of TNBC tumor cells using GSDensity". Maybe to better show the utility of GSDensity could be helpful to apply it to a bigger breast cancer dataset compared to single-cell data collected from more than 20 breast cancer patients like <https://www.embopress.org/doi/full/10.15252/embj.2020107333> or on the single cell breast cancer cell line dataset here <https://www.nature.com/articles/s41467-022-29358-6>
6. Paragraph "GSDensity identifies common spatially relevant pathways in six cancer ST datasets". The authors state in lines 318-320, "We observed that the BIOCARTA granulocyte pathway displayed an interesting pattern that generally surrounded the tumour cells" This must be quantified a showed to be statistically significant. Given the spatial information, this analysis can be easily performed.

Reviewer #3 (Remarks to the Author):

In this article, the authors presented a new way of analyzing scRNA-seq and spatial transcriptomics data mainly using:

1. MCA to construct a common low dimensional space of cells and genes.
2. Based on the low dimensional graph, each cell can be assigned with a pathway score by the network propagation step.
3. By testing the density of pathway score weighted spatial map, pathways with spatial specificity can be further retrieved.

The authors have evaluated the performance of their method and some of the similar methods proposed previously using simulations. And GSDensity outperformed most of the methods.

In a breast cancer study, GSDensity was able to fetch a subgroup of actively dividing cells, which is highly associated with the high expression of one receptor TYRO3, suggesting a potential role of TYRO3 in cancer cell proliferaton. This set of actively dividing cells could not be detected using any other clustering based method.

In another spatial transcriptomic (ST) study in mouse brain, GSDensity also identifies the stage related pathway along the trajectory path. And by merging the GSDensity predicted pathway scores with the spatial map, the authors were able to identify both spatial specific pathways and cluster specific pathways. Using other cluster centric method, people can identify cluster specific pathways. However only GSDensity is able to identify spatial relevant pathways.

In the last study, different ST data from several tumor types were collected. Using GSDensity, the authors found a common feature in six cancer types. Cells with active granulocyte pathway are surrounding the tumor clones. This common feature is important for biologist to design common therapeutic target for a wide range of cancer diseases.

Overall, this is a very creative and inspiring work to follow. It is always hard to choose the cluster numbers and hard to align clusters from multiple scRNA-seq/ST studies. GSDensity could overcome these challenges from another angle.

Some major comments:

1. In Extended Data Fig.1d, authors use size matched marker gene sets to calculate the accuracy of GSDensity. But in reality, the marker gene sets always have different size ranging from tens to hundreds. Whether the same accuracy can be achieved in real data?
2. MCA method can only be applied to categorical data. The binarization of the expression matrix will somehow loss the count nature of the data. Because the transcription is a dynamic process, when a cell is in the pulsing stage, the observed gene expression is close to zero, even though the gene is actively transcribed. I do not think MCA has the potential to model such bursting events.
3. GSDensity can be regarded as a cell-annotation method, and its annotation performance for cells of interest becomes particularly intriguing when compared with established annotation methods like scSorter and clustering+markers annotation methods. In the context of scRNA-seq data, a comparison can be made with Louvain clustering; whereas, in the case of spatial transcriptomics data, DR-SC clustering can serve as a relevant point of comparison.

Minor comments:

Some of the figures have weird characters, such as fig.3a-d, and Extended Data Fig.1b do not have any plot inside. Please refine the relevant figures.

We appreciate all the efforts and comments from reviewers. Our point-to-point reply is in blue. The line numbers refer to the ones in the Word file in 'simple markup' mode.

REVIEWER COMMENTS

Reviewer #1 (Remarks to the Author):

The authors presented GSDensity, which is a pathway-centric analysis method for scRNA-seq and spatial transcriptomics data. Using the MCA method to project cells and genes to a common latent space, GSDensity finds cells that are associated with a given gene set. The manuscript includes results found by this method which are potentially biologically meaningful. Overall, the idea of using gene sets to find relevant cell sets is interesting and a method with this purpose can be a valuable addition to the current single cell methods. The manuscript, however, should be improved:

1. From the description of the methods and the demonstrated use cases, GSDensity takes a given gene set, evaluates whether it has true coordination, and finds cells for which these genes are major contributors of their profiles and possibly functions. It appears that users need to pre-define a set of candidate gene sets, and evaluate and analyze them one by one. GSDensity does not provide gene sets. The abstract should make this point clear. The current wording like “detect biologically distinct gene sets” and “identify pathways that are active at various stages” could also mean that GSDensity finds such gene sets of pathways from data.

The reviewer is correct that users of GSDensity need to pre-define gene sets of interest. Users can curate gene sets based on their domain knowledge or prior experimental results. Also, as we showed in this manuscript, users can use publicly curated gene sets such as GO and KEGG. We revised our abstract to avoid the misunderstanding (line 18-24):

*“Using predefined functional gene sets, we show that GSDensity can not only accurately detect biologically distinct ones but also reveal novel cell-pathway associations that are ignored by existing methods. This is particularly evident in characterizing cancer cell states that are transcriptomically distinct but are driven by shared tumor-immune interaction mechanisms. Moreover, we show that GSDensity, combined with trajectory analysis can identify **curated** pathways that are active at various stages of mouse brain development.”*

2. The manuscript includes results from different datasets and cell types, and different initial gene sets or pathways were used as input to GSDensity to test. It would be helpful if the authors can provide guidance on how to obtain the initial sets of pathways to test, and ideally provide a generic set of pathways for users to use as default.

We agree this is a good idea and we added some guidance accordingly (line 458-468):

“Our original motivation of developing GSDensity is to build a tool for ‘pathway-centric’ analysis, with which we can dive into the data directly from the angle of pathways of interest. Initial gene sets could come from any prior knowledge, information, or preliminary data, such as RNA-seq data, CRISPR screen results, or GWAS genes, etc. When there lacks a prior knowledge regarding which specific pathways to test, users can start with carefully curated pathways in the public databases. For example, in the E17.5 mouse example, we used KEGG and BIOCARTA but not GO Biological Processes, because the latter included many cell type-related terms, while the mouse cells are largely un-differentiated at that stage. We chose to use GO Biological Processes for adult mouse brain ST data. For cancer ST data, since we aimed at finding common spatially relevant programs from multiple datasets, we chose to broaden our hypothesis space by using all gene sets from GO, Hallmarks, BIOCARTA, and KEGG.”

3. Some terms used in the manuscript need further clarification or definition. For example, in the Introduction, it's not clear what “heterogeneity of a pathway” means. Also, the “coordination” of a gene set is not a term that is commonly used.

We appreciate this comment. By “heterogeneity of a pathway”, we actually meant pathway activity variation across cells. For “coordination”, we agree that this is not a standard term in this field and thus requires better definition before using it. We have revised the manuscript accordingly for both cases (line 63-64):

“GSDensity uses multiple correspondence analysis (MCA) to co-embed cells and genes into a latent space and quantifies the overall variation of pathway activity levels across cells by estimating the density of the pathway genes in the latent space.”

(line 83-86):

“In the context of scRNA-seq data, considering a curated functional gene set, when the genes from this gene set are highly and specifically expressed in a subpopulation of the cells, we call such a gene set ‘coordinated’. This subpopulation of cells is thus defined as highly relevant cells to this gene set. A randomly selected list of genes is anticipated to be of low coordination, with no relevant cells.”

4. MCA constitutes the main part of the method of GSDensity, and the success of GSDensity in identifying cells most relevant to a set of genes relies on that the distance between cell embedding and gene embedding is meaningful, in the sense that the cell embedding and gene embedding are comparable. In addition to giving the MCA calculation process which is included in Methods, the authors are suggested to discuss and justify that the cell embedding and gene embedding are comparable.

This is a good point. We have discussed and justified this point in the revised the manuscript accordingly. We demonstrated that the distribution of distances across different entities (cells or genes) in the MCA embeddings are numerically comparable, because the coordinates of the entities are mutually dependent (generated from the same cell-gene matrix). The distance between a gene and a cell generally reflects how specifically the gene is expressed in the cell. Also, as the reviewer pointed out in Comment 5, the neighborhood size is an important parameter in our design, and we have experiments showing that GSDensity is robust to different choices of this parameter in a wide range. We revised the manuscript as follows (line 376-385):

“Although varying by datasets, the cell embeddings and the gene embeddings produced by the MCA are largely comparable, as shown by the overlapping distance distributions in multiple real-world scRNA-seq datasets (Supplementary Fig.1). In the MCA biplots, a cell-gene distance reflects the mutual specificity between a gene and a cell. The coordinates of the genes (or cells) are determined by the coordinates of the associated cells (or genes) weighted by their mutual specificity. The information of ‘seed’ genes can then be amplified by similar cells through highly specific cell-gene pairs. This design would make the size of neighborhood an important parameter when constructing the cell-gene graph. In our experiments, we found that the performance of GSDensity is robust and stable with respect to this parameter (Supplementary Figs. 2-4, Methods).”

5. GSDensity constructs a nearest neighbor graph as a key step in its pipeline. The size of the neighborhood is an important parameter. It’s important to show if the results are sensitive to this parameter and how should users set this parameter in practice.

We agree the number of neighbors to search is an important parameter. A short answer is that we suggest using 300 as the default parameter and the performance of GSDensity does not change much when it is set from 200 to 600. In the revised manuscript, we performed additional experiments (Supplementary Figs 2-4) and briefly described the results in Methods section. We also revised the writing to include more details and interpretations regarding these results (line 540-545):

“The default number of neighbors is set to be 300 and can be tuned by users. We have evaluated the robustness of GSDensity performance to this parameter in terms of pathway activity in the PBMC data (Supplementary Fig 2) and in terms of cell type classification in the 8 real-world datasets (Supplementary Figs 3-4). We found that, consistently across datasets, the performance of GSDensity increases as the number of neighbors increases from 20 to 200 and reaches plateau afterwards.”

6. In Fig. 4b, it seems that 20 partitions are used. How are these partitions obtained? Are these the clusters shown in Extended Data Fig. 8b? It is confusing because Data Fig. 8b has 12 clusters instead of 20.

We appreciate that the reviewer pointed this out. In Extended Data Fig. 8b, we were showing the clusters retained for the pseudotime calculation (Figure 4a). These clusters were results from unsupervised clustering from the default Seurat pipeline.

In Figure. 4b, as we already calculated the pseudotime scores, we partitioned the cells equally into 20 partitions based on their pseudotime scores. We aggregated the pathway activity level (PAL) within each partition to obtain a more robust estimation of pathway activities along the trajectory. This is for the purpose of clustering the pathways based on their patterns along the trajectory with a higher robustness.

We revised our manuscript to clarify (line 745-748): *“We equally split the cells into 20 partitions along the pseudotime trajectory based on the inferred pseudotime for the purpose of quantifying pathway activity along trajectory. These partitions were not related to the original clustering shown in Extended Data Fig.8a-b.”*

Minor points:

- In Fig. 2, please use uppercase and lowercase letters properly for the methods' names.
We revised the figure accordingly. We also revised all other figures with the same issue.

- Fig. 3i was referred to in the manuscript but the plot was not labeled separately.
This was a mistake and we meant Fig. 3h. We revised the manuscript accordingly. (Line 220)

- Line 97 should refer to the Methods section here.

We referred this sentence to Methods. (line 97-99)

"We randomly sample multiple size-matched gene sets and compute differential density in the same way (Fig.1c, middle), and the resulting differential density levels are used to generate a null distribution for estimating statistical significance (Methods)."

Reviewer #2 (Remarks to the Author):

In this manuscript, the authors proposed GSDensity, a computational method that uses multiple correspondence analysis (MCA) to reconstruct pathway activity at the single-cell level. Although pathway analysis using single-cell datasets is not a brand-new topic, the author compared their method with other state-of-the-art methods on several simulated and real datasets. They show that their method usually performs similarly or better than CelliD, another popular method to perform pathway analysis on single-cell data and based on multiple correspondence analysis like GSDensity. Although interesting, the manuscript is more challenging to follow in some parts (particularly the paragraph on TNBC) that should be rewritten to be more straightforward. Here are some comments that I believe could help in improving the manuscript:

1. Scalability is a crucial issue when it comes to single-cell analysis. However, no details about computing time are presented in the article. Can the authors deepen this aspect and compare their method with the others in terms of average running time at increasing number of considered cells (i.e. from 10,000 to 50,000 or more) and pathways? This analysis could also help the authors show an additional advantage of their method compared to celliD, which performs similarly on many occasions.

We agree with the point. In response, we performed runtime benchmarking experiments among GSDensity and several other pathway analysis methods. We found that CelliD was the fastest method while GSDensity ranked the second. For large datasets (40k cells), GSDensity is at least twice faster than any methods except for CelliD. Here are the results in detail, elaborated in the revised manuscript (line 427-436):

"The runtime for the MCA step with different number of cells were tested (Supplementary Figure 7a). It appeared increasing linearly with respect to the number of cells. The runtime for gene set coordination testing (without considering the MCA embedding calculation time) is more related to the number of gene sets instead of number of cells (Supplementary Figure 7b). We further compared the pathway activity scoring speed among five methods: GSDensity, CelliD, AUCell, ssGSEA, and VAM (Supplementary Figure 7c). In general, CelliD was the fastest, followed by GSDensity, AUCell, VAM, and ssGSEA. AUCell was faster than GSDensity when the number of cells was small while GSDensity was more scalable. GSDensity could calculate the pathway activity for 1000 gene sets in a dataset with 40,000 cells in about 10 minutes while AUCell took about 27 minutes."

Methods:

Line (703-713): *"For benchmarking the runtime of GSDensity and other methods, we used the public dataset 'hcbm40k' (SeuratData R package) which has 40,000 cells. We randomly selected gene sets from the Gene Ontology Biological Processes database, as input to the methods. We dissected GSDensity into three parts for runtime benchmarking: calculating MCA embeddings (Supplementary Fig. 7a), testing gene set coordinations (Supplementary Fig. 7b), and calculating pathway activities (Supplementary Fig. 7c). We compared the pathway activity calculation speed between GSDensity and several other methods. For all the methods, preprocessing steps were not recorded in the runtime benchmarking, such as the MCA calculation for GSDensity and CelliD, the pre-calculation of gene rankings for AUCell, etc. The runtime recording was performed using the R package 'microbenchmark'. The runtime benchmarking was performed using Linux (Redhat Enterprise Linux) system, with 12 cores, 120G RAM."*

Supplementary Figure 7. Runtime for GSDensity and other pathway analysis methods.

a. Runtime for MCA embedding calculation with different number of cells as the input.

b. Runtime for gene set coordination test with different number of cells and gene sets as the input.

c. Runtime for pathway activity scoring in five methods. For visualization purpose, the y-axis was log-transformed.

2. In GSDensity, the authors apply MCA to data normalized with Seurat. Can the normalization affect the results of the enrichment analysis? Can the authors try to decompose the data matrix resulting from a different normalization procedure (i.e. $\log(\text{CPM}+1)$ or scanpy normalization and so on)?

We agree this is a good question. GSDensity currently uses the default Seurat normalization, which normalizes each count by the total counts of that cell, multiply by a scale factor (10000), add a pseudo-count (1), and transform that value by natural logarithm. The default normalization of Scanpy ('scanpy.pp.normalize_total' and 'scanpy.pp.log1p') uses the same method. This was also used by CellID for calculating the MCA step. To our knowledge, besides the Seurat default normalization, there are two other normalization strategies being regularly used: the SCTransform method (by Seurat) and the Scrans method. Thus, we compared the GSDensity results between the Seurat default and these two other methods and found that the results were largely comparable. In our revised manuscript (line 450-456, Supplementary Figure 9):

"We also tested whether the performance of GSDensity will be affected by different normalization and transformation strategies for scRNA-seq data by comparing three strategies: the default Seurat normalization (also used by GSDensity by

default), the SCTransform⁶² implemented by Seurat, and the scran⁶³ strategy offered by the scran R package. We found that these normalization strategies gave highly consistent results for both gene set coordination test (Supplementary Figure 9a-c) and gene set PAL calculation (Supplementary Figure 9d).

Supplementary Figure 9. Performance of GSDensity using different normalization strategies for scRNA-seq data. a-c. Correlation of gene set coordination calculation between default Seurat ('default') and SCTransform ('SCT') (a), default and scran (b), and SCT and scran (c). We used the PBMC3K data. Negative log-transformed p-values were used as the metric of gene set coordination. The first 2,000 GO Biological Processes gene sets were used, with 1,751 passing the filter of at least 3 genes from the list being detected.

d. Correlation of gene set pathway activity levels (PALs) at single cell levels. For the gene sets used in a-c, 224 showed strong coordination ($p\text{-value} < 0.01$). PALs were calculated in the PBMC3K datasets with different normalization strategies. Pearson's correlation coefficients were calculated for each of the 224 gene sets comparing default and scran, default and SCT, and scran and SCT."

3. Another important noisy aspect of single-cell data is the batch effect. Can the others evaluate how batch affects pathway activity? For this, the authors can use many available datasets in the literature. See, for example, <https://www.nature.com/articles/s41592-019-0619-0> or <https://www.nature.com/articles/s41592-018-0254-1>

We agree that batch effect could introduce noise into single-cell data. GSDensity does not have its own function to tackle batch effects. Per the design of GSDensity, as we mentioned, it is recommended to use batch correction algorithms such as ComBat, Scanorama, etc., to obtain corrected gene expression prior to using GSDensity. We showed an example that GSDensity performed better in batch-corrected data (ComBat) than uncorrected. Here is the revised manuscript (line 443-450, Supplementary Figure 8):

"For integrated datasets from multiple origins with strong batch effects, since the MCA uses the original gene expression data, the batch effect correction should also be performed on the original gene expression data, instead of on the low-dimensional embeddings, using methods such as ComBat⁶⁴, CarDEC⁶⁵, or Scanorama⁶⁶, etc. We showed that although

GSDensity performed reasonably well without having batch correction, using ComBat to adjust the gene expression data can improve its performance (Supplementary Fig.8). Proper batch correction improves the performance of GSDensity, while over- or under- correction may introduce bias. Thus, we suggest to carefully evaluate batch correction outcome before performing pathway analysis.”

Supplementary Figure 8. Performance of GSDensity in data with batch effect before and after batch correction. a-b. UMAP demonstration of a human pancreas dataset (uncorrected). The cells were colored by annotated cell types (a) or sequencing technology (b). c-d. UMAP demonstration of a human pancreas dataset (corrected). The cells were colored by annotated cell types (c) or sequencing technology (d). e. The cell identity recovery using marker genes (AUC metric) in batch-corrected and uncorrected data. We used full marker sets and marker sets mixing with 1:1 and 1:5 random genes.”

4. It Would be nice to add comparison with at least another method based on low dimensionality spaces like the recently published scGSEA you can find here <https://academic.oup.com/nargab/article/5/1/lqad024/7069282>

We added scGSEA to our benchmarking and it generally performed well, especially with the ACC metric. We think in general, GSDensity, CelliD, and scGSEA were the best performing methods among those benchmarked. Here we refer to our updated figures:

Fig. 2: Benchmarking the GSDensity method.

a. An illustration of gene set density in pbmc3k data using all genes (top left), B cell markers (top right), reduced B cell markers with random genes (bottom left) and randomly sampled genes (bottom right). Contours are plotted to visualize the density of gene sets, which is estimated using the UMAP space. It is worth notice that in the real data analysis, GSDensity directly estimate the density of gene sets in the MCA space, not the UMAP space.

b. Schematic of the gene sets used for panel c-e. Marker gene lists were first collected for each cell type. The 'all marker sets' were generated by randomly selecting 80% of the marker genes and such randomizations were repeated 5 times for each marker list. The 'marker mix1 sets' were synthesized by randomly selecting 40% of the marker genes and the same

number of randomly genes. The 'marker mix3 sets' were synthesized by randomly selecting 20% of the marker genes and three-times the number of random genes. Similarly, the randomizations for 'mix1' and 'mix3' sets were performed 5 times for each marker list. Lastly, random gene sets were synthesized by randomly select genes with the same numbers and sizes as the previous three types of gene sets.

c-e. Validation of the sensitivity of GSDensity to identify gene sets with coordination. Cell type markers ('all.marker.set'), markers with size-matched random genes ('marker.mix1.set'), markers with three-folds size-matched random genes ('marker.mix3.set'), and all random genes ('random.set'), were used as input gene sets to calculate their coordination in the pbmc3k (c), lung (d), and pancreas (e) data, respectively. The red dashed line showed the unadjusted p-value equal to 0.05. The center line of the box plot showed the median of data; the box limits showed the upper and lower quartiles; the whiskers showed 1.5 times interquartile range and points showed outliers.

f. Benchmarking the reliability of gene set scoring aspect of GSDensity and six popular tools on simulated datasets. Each row represents a method, and each column represents the gene set and dataset condition (average of three parallel datasets of the same mode and noise level). The colors and the sizes of the dots both demonstrate the AUC score. The colors and the sizes of the dots both demonstrate the AUC score. The color scale was forced the same for all the three sub-panels for cross-dataset comparison. The size scale was automatically decided for best demonstrating the within-dataset contrast.

g. Benchmarking the reliability of gene set scoring aspect of GSDensity and six popular tools on real datasets. Cell type markers were first used with their original sizes, and then got their specificity decreased by reducing the size to 70% and 30%, and further by mixing with random genes (one-fold, three-folds, and five-folds). The colors and the sizes of the dots both demonstrate the AUC score. The color scale was forced the same for all the three sub-panels for cross-dataset comparison. The size scale was automatically decided for best demonstrating the within-dataset contrast.

Extended Data Fig.4. Using the ACC score to evaluate the reliability of gene set scoring of GSDensity and six popular tools on simulated datasets and real-world datasets. Each row represents a method, and each column represents the gene set and dataset condition. For simulated datasets, the columns represent the average of three parallel datasets of the same mode and noise level. The colors and the sizes of the dots both demonstrate the ACC score. The colors and the sizes of the dots both demonstrate the ACC score.

Extended Data Fig.5. Benchmarking the reliability of gene set scoring of GSDensity and six popular tools on real-world datasets (related to Fig.2g), using the AUC score. Each row represents a method, and each column represents the gene set condition. The colors and the sizes of the dots both demonstrate the AUC score. The colors and the sizes of the dots both demonstrate the AUC score.

5. Paragraph “Classification of TNBC tumor cells using GSDensity”. Maybe to better show the utility of GSDensity could be helpful to apply it to a bigger breast cancer dataset compared to single-cell data collected from more than 20 breast cancer patients like <https://www.embopress.org/doi/full/10.15252/emj.2020107333> or on the single cell breast cancer cell line dataset here <https://www.nature.com/articles/s41467-022-29358-6>

We agree it is a good idea to add another cohort of TNBC samples into our analysis. We analyzed the 8 TNBC data from this report <https://www.embopress.org/doi/full/10.15252/emj.2020107333>. We found that TYRO3 was not highly expressed in 7 of the samples (in our previously analyzed cohort, TYRO3 was not highly expressed in 3 out of 5 samples) but for the one sample TYRO3 highly expressed, it showed higher expression in the actively dividing tumor cells detected by GSDensity, with p-value 0.039. We think this observation could be related to the inter-patient heterogeneity of tumor transcriptome. Similar to the conclusion drew from the previously analyzed cohort, our analysis postulated a possible role of GAS6-TYRO3 signaling in TNBC tumor proliferation that exist in a portion of the patients. We did not follow the suggestion of using cell line dataset because these cell lines are all immortalized cells, and they are generally actively dividing. We confirmed this by checking the expression of MKI67 through the interactive portal provided by authors (https://bcAtlas.tigem.it/tigem/dibernardo/AIRC_atlas_32_ccls/?ds=Atlas_32_ccls).

Here we refer to the revised manuscript (line 224-237):

“The high expression of TYRO3 in actively dividing cells was also observed in the TNBC-5, confirming the previous finding (p-value $3.18e-8$, Wilcoxon test, Extended Data Fig.6l-m). We then investigated this TYRO3 expression pattern in another published cohort with 8 TNBC patient samples³⁰. TYRO3 were lowly detected in 7 of the samples (detected in 1%-8% of tumor cells). In the only sample (GSM4909284_TN-MH0114-T2) with relative high expression of TYRO3 (detected in 24% of tumor cells), the actively dividing cells showed higher expression of TYRO3 than other tumor cells (p-value 0.039, Wilcoxon test, Extended Data Fig.6n). These results indicated that the overall expression level of TYRO3 in breast cancer cells is highly patient specific, while the high-TYRO3 expressing samples always had TYRO3 preferably express in a small group of actively dividing cells. The GAS6-TYRO3 axis has been associated with tumor cell proliferation, malignancy, and anti-PD1/PD-L1 resistance in previous studies³¹⁻³⁵. Thus, through the integration of data and prior

knowledge using GSDensity, we postulated a potential role TYRO3 in TNBC proliferation using only a few TNBC samples with very sparse single-cell gene expression profiles and generated a testable hypothesis for further studies.”

6. Paragraph “GSDensity identifies common spatially relevant pathways in six cancer ST datasets”. The authors state in lines 318-320, “We observed that the BIOCARTA granulocyte pathway displayed an interesting pattern that generally surrounded the tumour cells” This must be quantified and showed to be statistically significant. Given the spatial information, this analysis can be easily performed.

We agree that a quantification and statistical test should be performed besides visual inspection/interpretation. For each of the six datasets, we extracted the number of tumor boundary cells and that of ‘positive’ cells for the pathways of interest, performed a Chi-squared test using a 2-by-2 contingency table. We found that the boundary cells enrich BIOCARTA granulocyte pathway relevant cells in PC, IC, OC, and PACC, but not BC or CC. We had similar findings for the mesenchyme morphogenesis pathway. We appreciate this comment from the reviewer that made our claim more solid. We revised the manuscript accordingly (line 330-344):

“We observed that the BIOCARTA granulocyte pathway displayed interesting patterns that generally surrounded the tumor cells, which were visualized as contours (Fig.6a-f, Extended Data Fig.13). This observation was statistically significant (Methods, Chi-square test) in the PC (p-value 1.575e-14), IC (p-value 0.014), OC (p-value 2.429e-12), and PACC (p-value 0.030), while not for BC or CC. This observation suggested granulocytic infiltration towards the tumor cells, which has been found to be associated with tumor progression and metastasis⁴³⁻⁴⁶. We also saw that the spots with highly active granulocyte pathway are not homogeneously surrounding all the tumor clones, such as those in the cervical cancer (Fig.6c) and in the prostate cancer samples (Fig.6f). We also noticed that the mesenchyme morphogenesis pathway from GO biological process terms, showed spatial relevance in all the datasets (Extended Data Figs.14-15). The data spots with high activity of this pathway overlapped with the tumor cells and appeared enriched at the borders of the tumor clones (Methods. for BC, p-value 1.805e-11; CC, p-value < 2.2e-16; IC, p-value < 2.2e-16; PACC, p-value 0.004). This is consistent with the current understanding that mesenchymal cells are highly related to tumor invasion^{47,48}.”

Methods in line 798-805:

“To perform enrichment analysis of certain pathways in tumor boundary cells, we partitioned the cells in the ST datasets in two ways. First, we partitioned the cells based on their relevance to the pathways of interest (granulocyte pathway and mesenchyme morphogenesis pathway in our cases) the same ways as demonstrated in the TNBC analysis (Methods). Second, we partitioned the cells based on whether they were boundary cells, which were defined as those with at least 1 and at most 4 of the neighbors being tumor cells, considering the tightly packing pattern of Visium data. We then performed Chi-square test to examine whether boundary cells enrich cells of the relevant pathways.”

Reviewer #3 (Remarks to the Author):

In this article, the authors presented a new way of analyzing scRNA-seq and spatial transcriptomics data mainly using:

1. MCA to construct a common low dimensional space of cells and genes.
2. Based on the low dimensional graph, each cell can be assigned with a pathway score by the network propagation step.
3. By testing the density of pathway score weighted spatial map, pathways with spatial specificity can be further retrieved.

The authors have evaluated the performance of their method and some of the similar methods proposed previously using simulations. And GSDensity outperformed most of the methods.

In a breast cancer study, GSDensity was able to fetch a subgroup of actively dividing cells, which is highly associated with the high expression of one receptor TYRO3, suggesting a potential role of TYRO3 in cancer cell proliferation. This set of actively dividing cells could not be detected using any other clustering based method.

In another spatial transcriptomic (ST) study in mouse brain, GSDensity also identifies the stage related pathway along the trajectory path. And by merging the GSDensity predicted pathway scores with the spatial map, the authors were able to identify both spatial specific pathways and cluster specific pathways. Using other cluster centric method, people can identify cluster specific pathways. However only GSDensity is able to identify spatial relevant pathways.

In the last study, different ST data from several tumor types were collected. Using GSDensity, the authors found a

common feature in six cancer types. Cells with active granulocyte pathway are surrounding the tumor clones. This common feature is important for biologist to design common therapeutic target for a wide range of cancer diseases.

Overall, this is a very creative and inspiring work to follow. It is always hard to choose the cluster numbers and hard to align clusters from multiple scRNA-seq/ST studies. GSDensity could overcome these challenges from another angle.

Some major comments:

1. In Extended Data Fig.1d, authors use size matched marker gene sets to calculate the accuracy of GSDensity. But in reality, the marker gene sets always have different size ranging from tens to hundreds. Whether the same accuracy can be achieved in real data?

We designed the accuracy score as a metric to compare the performance of GSDensity and other methods. We used size-matched marker sets through downsampling to control for the uncertainty introduced by the uneven size of markers. We agree with the reviewer, and this is an important question especially if a user considers using GSDensity as a cell type annotation tool. To assess the performance of GSDensity in competitively annotating cell types with markers in full sizes, we calculated the accuracy scores using the 27 simulated datasets and compared the results to the ones we got through the benchmarking experiments (Extended Figure 4a-c). We found that in most scenarios, the full marker lists, although sizes could vary by folds, allowed GSDensity to perform better than using the truncated marker lists. With this result, we recommend users to using a full marker set for such tasks. We added this result to our manuscript (line 411-416):

“GSDensity can also be applied as a cell type annotator when marker genes were available. We showed that GSDensity performed best among pathway analysis tools in this task (Extended Data Figure 4). For the benchmarking experiment, we used randomly truncated marker gene sets to control the uncertainty caused by variable gene set sizes. We further found that the performance of GSDensity in cell type prediction would increase when using full marker lists (with variable sizes) in most of the scenarios introduced in the 27 SERGIO-simulated datasets (Supplementary Fig.5).”

Supplementary Figure 5. GSDensity performance on cell identity prediction with truncated (to 10) marker sets and full marker sets (length varies) using simulated data (Mode-1, a; Mode-2, b; Mode-3, c). For each mode, the data with three different sparsity levels were investigated combined with marker sets with different signal strengths (strong, medium, and weak). The data with truncated marker sets were the same as that used in Extended Data Figure 4a-c.

2. MCA method can only be applied to categorical data. The binarization of the expression matrix will somehow lose the count nature of the data. Because the transcription is a dynamic process, when a cell is in the pulsing stage, the observed gene expression is close to zero, even though the gene is actively transcribed. I do not think MCA has the potential to model such bursting events.

We generally agree with the reviewer. Gene transcription is indeed a dynamic process. From our experience, in most cases (genes), the sparse and noisy scRNA-seq data does not capture the continuity nature of various gene expression levels in a cell population, with sufficient dynamic range of transcript detection. We agree that at some point, when this type of continuity is captured, the binarization step would cause loss of information. The binarization used by MCA can be seen as a denoising step to some extent and we agree it does have trade-offs in reducing technical noises and losing biological signals.

3. GSDensity can be regarded as a cell-annotation method, and its annotation performance for cells of interest becomes particularly intriguing when compared with established annotation methods like scSorter and clustering+markers annotation methods. In the context of scRNA-seq data, a comparison can be made with Louvain clustering; whereas, in the case of spatial transcriptomics data, DR-SC clustering can serve as a relevant point of comparison.

Although not designed as a cell-annotation method, we agree that GSDensity can be used for the purpose. We compared GSDensity with scSorter in the simulated datasets. We also derived a clustering+marker strategy and added that to the comparison. We used the simulated datasets because they covered different scenarios such as marker signal strengths, data sparsity, and cell type dynamics. It also introduced less bias when comparing cluster-free methods (GSDensity and scSorter) against cluster-based methods since many ground-truth from the real-world datasets were derived using cluster-based strategies. We found that the cluster-based strategy performed best in scenarios with stable cell types while GSDensity performed best in scenarios with dynamic cell states. Here are more detailed description and interpretation of these experiments (line 416-423):

“We also compared the performance of GSDensity with another marker-based cell type annotation tool, scSorter⁶² and a clustering-based strategy for automatic cell annotation (Methods), using the 27 SERGIO-simulated datasets (Methods, Supplementary Figure 6). We noticed that GSDensity and the clustering-based strategy generally out-performed scSorter. In the scenarios with stable cell types, the clustering-based strategy performed better than GSDensity. In the scenarios with dynamic cell states, especially as the data sparsity increases, GSDensity showed superior performance over the clustering-based strategy. It is worth noting that with an increase in cellular throughput, datasets with sparsity close to or higher than 90% became very common in recent single-cell studies.”

Methods in line 808-816:

“We applied scSorter⁶² on the 27 SERGIO-simulated datasets, using the same marker sets that we used for benchmarking GSDensity with other pathway scoring tools. Default settings were used as described here: <https://cran.r-project.org/web/packages/scSorter/vignettes/scSorter.html>. We also employed a ‘cluster plus marker’ strategy (Supplementary Figure 6) for cell type annotation. For each dataset, we first used Louvain clustering (resolution 1.0) to cluster cells. We then use the “AddModuleScore” function to calculate the normalized average expression of each marker set in single cells and calculate the average marker expression level for each cluster. After scaling, we annotate the cluster based on the highest average marker expression level. The ACC metric was the same as described above to evaluate the method performance.”

Supplementary Figure 6. Comparing the cell type annotation accuracy among GSDensity (green), scSorter (blue), and a cluster-based strategy (red, Methods) using simulated data (Mode-1, a; Mode-2, b; Mode-3, c). For each mode, the data with three different sparsity levels were investigated combined with marker sets with different signal strengths (strong, medium, and weak). For all three methods, full marker sets were used.

Minor comments:

Some of the figures have weird characters, such as fig.3a-d, and Extended Data Fig.1b do not have any plot inside. Please refine the relevant figures.

We appreciate the comments. We changed the character fonts in some figures including fig.3a-d.

As for Extended Data Fig.1b, we did have plot from our end as below:

We did re-insert this figure in our manuscript file and hopefully this time it showed up as expected.

Reviewer #1 (Remarks to the Author):

All my concerns have been addressed in the revised manuscript.

Reviewer #2 (Remarks to the Author):

The authors have addressed all my concerns, providing comprehensive details in their responses. However, I'd like to emphasize one final crucial point that should be addressed before the publication of the manuscript. To enhance the manuscript's clarity, I propose including two additional panels, Panel H and Panel I, within Figure 2. These panels would serve to summarize the performance of the seven tested methods across all the simulated and real datasets utilized in their evaluation.

To implement this suggestion, the authors can employ boxplots to visually represent the distribution of estimated AUC (Area Under the Curve) values for each method across specific types of datasets (i.e., real or simulated). Specifically:

Novel Panel H: On the x-axis, the author should arrange the seven methods according to the median AUC values derived from all simulated datasets on which the methods were tested. It's important to note that this should encompass the three datasets displayed in Panel F of Figure 2 and the datasets featured in Extended Figure 4.

Novel Panel I: It should be like novel panel H but using AUC values from all real datasets on which the methods were tested. This should also encompass datasets from Extended Figure 4 and Figure 5.

Adding these two panels in Figure 2 will significantly enhance the manuscript's comprehensibility and provide readers with a more holistic view of the methods' performance across a broader range of datasets. These results can be used to refine the discussion paragraph better if necessary. This last refinement will contribute to the overall quality and clarity of the publication.

Reviewer #3 (Remarks to the Author):

There are several additional points we want to clarify with the authors:

1. The authors should discuss more of the binarization step. Is there a need to check the data quality before applying the binarization? in what scenario the binarization will produce unreliable or misleading outputs not reflecting the true biological process.
2. It is good to know that GSDensity has better annotation performance than the cluster-based method when dealing with dynamic cell states in scRNA-seq. This is an important guidance for readers to consider when choosing the annotation method for their own analysis. How about the performance in spatial transcriptomics? Can you further make a comparison between GSDensity and DR-SC (<https://github.com/feiyong/DR.SC>)

We appreciate all the efforts and comments from reviewers. Our point-to-point reply is in blue. The line numbers refer to the ones in the Word file in 'simple markup' mode.

REVIEWER COMMENTS

Reviewer #1 (Remarks to the Author):

All my concerns have been addressed in the revised manuscript.

We appreciate the comments from the reviewers that helped improve our manuscript.

Reviewer #2 (Remarks to the Author):

The authors have addressed all my concerns, providing comprehensive details in their responses. However, I'd like to emphasize one final crucial point that should be addressed before the publication of the manuscript. To enhance the manuscript's clarity, I propose including two additional panels, Panel H and Panel I, within Figure 2. These panels would serve to summarize the performance of the seven tested methods across all the simulated and real datasets utilized in their evaluation.

To implement this suggestion, the authors can employ boxplots to visually represent the distribution of estimated AUC (Area Under the Curve) values for each method across specific types of datasets (i.e., real or simulated). Specifically:

Novel Panel H: On the x-axis, the author should arrange the seven methods according to the median AUC values derived from all simulated datasets on which the methods were tested. It's important to note that this should encompass the three datasets displayed in Panel F of Figure 2 and the datasets featured in Extended Figure 4.

Novel Panel I: It should be like novel panel H but using AUC values from all real datasets on which the methods were tested. This should also encompass datasets from Extended Figure 4 and Figure 5.

Adding these two panels in Figure 2 will significantly enhance the manuscript's comprehensibility and provide readers with a more holistic view of the methods' performance across a broader range of datasets. These results can be used to refine the discussion paragraph better if necessary. This last refinement will contribute to the overall quality and clarity of the publication.

We agree that adding extra plots to summarize the benchmarking result is a good idea, and we put Figure 2h-i as suggested by the reviewer:

Figure and legend:

Fig. 2: Benchmarking the GSDensity method.

a. An illustration of gene set density in pbmc3k data using all genes (top left), B cell markers (top right), reduced B cell markers with random genes (bottom left) and randomly sampled genes (bottom right). Contours are plotted to visualize the density of gene sets, which is estimated using the UMAP space. It is worth notice that in the real data analysis, GSDensity directly estimate the density of gene sets in the MCA space, not the UMAP space.

b. Schematic of the gene sets used for panel c-e. Marker gene lists were first collected for each cell type. The 'all marker sets' were generated by randomly selecting 80% of the marker genes and such randomizations were repeated 5 times for each marker list. The 'marker mix1 sets' were synthesized by randomly selecting 40% of the marker genes and the same number of randomly genes. The 'marker mix3 sets' were synthesized by randomly selecting 20% of the marker genes and three-times the number of random genes. Similarly, the randomizations for 'mix1' and 'mix3' sets were performed 5 times for each marker list. Lastly, random gene sets were synthesized by randomly select genes with the same numbers and sizes as the previous three types of gene sets.

c-e. Validation of the sensitivity of GSDensity to identify gene sets with coordination. Cell type markers ('all.marker.set'), markers with size-matched random genes ('marker.mix1.set'), markers with three-folds size-matched random genes ('marker.mix3.set'), and all random genes ('random.set'), were used as input gene sets to calculate their coordination in the pbmc3k (c), lung (d), and pancreas (e) data, respectively. The red dashed line showed the unadjusted p-value equal to 0.05. The center line of the box plot showed the median of data; the box limits showed the upper and lower quartiles; the whiskers showed 1.5 times interquartile range and points showed outliers.

f. Benchmarking the reliability of gene set scoring aspect of GSDensity and six popular tools on simulated datasets. Each row represents a method, and each column represents the gene set and dataset condition (average of three parallel datasets of the same mode and noise level). The colors and the sizes of the dots both demonstrate the AUC score. The colors and the sizes of the dots both demonstrate the AUC score. The color scale was forced the same for all the three sub-panels for cross-dataset comparison. The size scale was automatically decided for best demonstrating the within-dataset contrast.

g. Benchmarking the reliability of gene set scoring aspect of GSDensity and six popular tools on real datasets. Cell type markers were first used with their original sizes, and then got their specificity decreased by reducing the size to 70% and 30%, and further by mixing with random genes (one-fold, three-folds, and five-folds). The colors and the sizes of the dots both demonstrate the AUC score. The color scale was forced the same for all the three sub-panels for cross-dataset comparison. The size scale was automatically decided for best demonstrating the within-dataset contrast.

h. Summary of benchmarking experiments with simulated data using the AUC metric. For each simulated dataset, the median of the AUC scores were used as a data point in this boxplot. The results were summarized for each method and grouped by different drop out levels. Methods were arranged along x-axis based on their average performance. For the boxplot, the center line of the box plot showed the median of data; the box limits showed the upper and lower quartiles; the whiskers showed 1.5 times interquartile range and points showed outliers.

i. Summary of benchmarking experiments with real data using the AUC metric. For each simulated dataset, the median of the AUC scores were used as a data point in this boxplot. The results were summarized for each method and grouped by different gene set selection strategies. Methods were arranged along x-axis based on their average performance. For the boxplot, the center line of the box plot showed the median of data; the box limits showed the upper and lower quartiles; the whiskers showed 1.5 times interquartile range and points showed outliers.

Revised main text (line 184-188)

"We summarize the performance of PAL scoring methods in simulated and real-world datasets in Fig.2h (simulated) and Fig. 2i (real-world), and we show that GSDensity has the most stable performance, as it shows more resistance to the increasing dataset sparsity or decreasing gene set specificity. In general, besides GSDensity, CelliD, scGSEA, and VAM showed reasonable performance."

Reviewer #3 (Remarks to the Author):

There are several additional points we want to clarify with the authors:

1. The authors should discuss more of the binarization step. Is there a need to check the data quality before applying the binarization? in what scenario the binarization will produce unreliable or misleading outputs not reflecting the true biological process.

We agree that it is worth discussion and revised our manuscript accordingly. We added the discussion below (line 387-398):

"With GSDensity, we also offer an option of PAL-based cell binarization for downstream analysis, as was demonstrated in the TNBC data analysis. Since the PAL calculation is an outcome of network propagation, for most coordinated gene sets (Fig. 1c), the PAL among all cells would have two or more modalities because the network propagation is restricted to the highly relevant cells and converges before propagating to less relevant cells. Thus, we recommend performing binarization on only gene sets that pass the coordination test. In our experience (e.g., PBMC data, GO Biological Process

gene sets), over 98% of coordinated pathways (1160 gene sets from the GO Biological Process collection) demonstrated more than one modalities (adjusted p-value less than 0.05, mode testing method by Ameijeiras-Alonso et al. using the R package 'multimode'). Occasionally, certain pathways can demonstrate more than two modalities. Thus, we recommend users to visually inspect the PAL distributions of important pathways to validate the automatic results and perform correction as necessary.”

2. It is good to know that GSDensity has better annotation performance than the cluster-based method when dealing with dynamic cell states in scRNA-seq. This is an important guidance for readers to consider when choosing the annotation method for their own analysis. How about the performance in spatial transcriptomics? Can you further make a comparison between GSDensity and DR-SC (<https://github.com/feiyong/DR.SC>)

We agree that GSDensity performs better on data with dynamic cell states than do clustering-based approaches. Clustering-based methods have been developed for spatial transcriptomics (ST) data leveraging both gene expression and spatial coordinates. As suggested, we compare marker-based cell annotation (GSDensity) and clustering-based strategy for ST data cell annotation, where the clustering was computed using DR.SC instead of Louvain clustering, on a real-world human tonsil ST data with author’s annotations. We found that GSDensity performed better than DR.SC. We added the following revisions (line 436-441):

“Additionally, we examined automatic cell annotation in spatial transcriptomics data comparing GSDensity with DR.SC, a clustering-based strategy encouraging spatial-smoothness (Methods). With the labeled human tonsil data, we found that GSDensity performed better than the clustering-based strategy (59.32% vs 55.69%; Supplementary Figure 7) indicating the importance of performing accurate molecular dissection in annotating ST data. “

Supplementary Figure 7. Comparing the cell type annotation accuracy between GSDensity (and a cluster-based strategy (Methods) in a spatial genomics dataset. We demonstrate the cell on 2-dimensional UMAPs with labels from original annotation (a), GSDensity-based prediction (b) and clustering-based prediction (c). Prediction of cells with GSDensity showing significant improvement were highlighted.

In method (line 326-335):

“We applied scSorter on the 27 SERGIO-simulated datasets, using the same marker sets that we used for benchmarking GSDensity with other pathway scoring tools. Default settings were used as described here: <https://cran.r-project.org/web/packages/scSorter/vignettes/scSorter.html>. We also employed a ‘cluster plus marker’ strategy (Supplementary Figure 6) for cell type annotation. For each dataset, we first used Louvain clustering (resolution 1.0) to cluster cells. We then use the “AddModuleScore” function to calculate the normalized average expression of each marker set in single cells and calculate the average marker expression level for each cluster. After scaling, we annotate the cluster based on the highest average marker expression level. The ACC metric was the same as described above to evaluate the method performance. For the ‘cluster plus marker’ strategy with spatial genomics data, we used DR-SC instead of Louvain clustering using default settings.”

Reviewer #2 (Remarks to the Author):

The Authors have addressed all my concerns

Reviewer #3 (Remarks to the Author):

Thanks for answering all my comments and I have no more questions.

REVIEWERS' COMMENTS

Reviewer #2 (Remarks to the Author):

The Authors have addressed all my concerns

We appreciate the comments of this reviewer as they helped to improve our study.

Reviewer #3 (Remarks to the Author):

Thanks for answering all my comments and I have no more questions.

We appreciate the comments of this reviewer as they helped to improve our study.